# Deep Reinforcement Learning Guided Improvement Heuristic for Job Shop Scheduling

**Cong Zhang[1], Zhiguang Cao[2], Wen Song[3,*], Yaoxin Wu[4] & Jie Zhang[1]**
[1]Nanyang Technological University, Singapore
[2]Singapore Management University, Singapore
[3]Institute of Marine Science and Technology, Shandong University, China
[4]Department of Industrial Engineering & Innovation Sciences, Eindhoven University of Technology
`cong030@e.ntu.edu.sg, yzgcao@smu.edu.sg, wensong@email.sdu.edu.cn`
`y.wu2@tue.nl, jzhang@ntu.edu.sg`

## Abstract

Recent studies in using deep reinforcement learning (DRL) to solve Job-shop scheduling problems (JSSP) focus on *construction* heuristics. However, their performance is still far from optimality, mainly because the underlying graph representation scheme is unsuitable for modelling partial solutions at each construction step. This paper proposes a novel DRL-guided *improvement* heuristic for solving JSSP, where graph representation is employed to encode complete solutions. We design a Graph-Neural-Network-based representation scheme, consisting of two modules to effectively capture the information of dynamic topology and different types of nodes in graphs encountered during the improvement process. To speed up solution evaluation during improvement, we present a novel message-passing mechanism that can evaluate multiple solutions simultaneously. We prove that the computational complexity of our method scales linearly with problem size. Experiments on classic benchmarks show that the improvement policy learned by our method outperforms state-of-the-art DRL-based methods by a large margin.

## 1 Introduction

Recently, there has been a growing trend towards applying deep (reinforcement) learning (DRL) to solve combinatorial optimization problems (Bengio et al., 2020). Unlike routing problems that are vastly studied (Kool et al., 2018; Xin et al., 2021a; Hottung et al., 2022; Kwon et al., 2020; Ma et al., 2023; Zhou et al., 2023; Xin et al., 2021b), the Job-shop scheduling problem (JSSP), a well-known problem in operations research ubiquitous in many industries such as manufacturing and transportation, received relatively less attention.

Compared to routing problems, the performance of existing learning-based solvers for scheduling problems is still quite far from optimality due to the lack of an effective learning framework and neural representation scheme. For JSSP, most existing learning-based approaches follow a dispatching procedure that constructs schedules by extending partial solutions to complete ones. To represent the constructive states, i.e. partial solutions, they usually employ disjunctive graph (Zhang et al., 2020; Park et al., 2021b) or augment the graph with artificial machine nodes (Park et al., 2021a). Then, a Graph Neural Network (GNN) based agent learns a latent embedding of the graphs and outputs construction actions. However, such representation may not be suitable for learning construction heuristics. Specifically, while the agent requires proper work-in-progress information of the partial solution during each construction step (e.g. the current machine load and job status), it is hard to incorporate them into a disjunctive graph, given that the topological relationships could be more naturally modelled among operations[1] (Balas, 1969). Consequently, with the important components being ignored due to the solution incompleteness, such as the disjunctive arcs among undispatched operations (Zhang et al., 2020) and the orientations of disjunctive arcs among operations within a machine clique (Park et al., 2021b), the partial solution representation by disjunctive graphs in

---

*corresponding author

[1]Please refer to Appendix K for a detailed discussion

current works may suffer from severe biases. Therefore, one question that comes to mind is: Can we transform the learning-to-schedule problem into a learning-to-search-graph-structures problem to circumvent the issue of partial solution representation and significantly improve performance?

Compared to construction ones, improvement heuristics perform iterative *search* in the neighbourhood for better solutions. For JSSP, a neighbour is a complete solution, which is naturally represented as a disjunctive graph with all the necessary information. Since searching over the space of disjunctive graphs is more effective and efficient, it motivates a series of breakthroughs in traditional heuristic methods (Nowicki & Smutnicki, 2005; Zhang et al., 2007). In traditional JSSP improvement heuristics, local moves in the neighbourhood are guided by hand-crafted rules, e.g. picking the one with the largest improvement. This inevitably brings two major limitations. Firstly, at each improvement step, solutions in the whole neighbourhood need to be evaluated, which is computationally heavy, especially for large-scale problems. Secondly, the hand-crafted rules are often short-sighted and may not take future effects into account, therefore could limit the improvement performance.

In this paper, we propose a novel DRL-based improvement heuristic for JSSP that addresses the above limitations, based on a simple yet effective improvement framework. The local moves are generated by a deep policy network, which circumvents the need to evaluate the entire neighbourhood. More importantly, through reinforcement learning, the agent is able to automatically learn search policies that are longer-sighted, leading to superior performance. While a similar paradigm has been explored for routing problems (Chen & Tian, 2019; Lu et al., 2019; Wu et al., 2021), it is rarely touched in the scheduling domain. Based on the properties of JSSP, we propose a novel GNN-based representation scheme to capture the complex dynamics of disjunctive graphs in the improvement process, which is equipped with two embedding modules. One module is responsible for extracting topological information of disjunctive graph, while the other extracts embeddings by incorporating the heterogeneity of nodes' neighbours in the graph. The resulting policy has linear computational complexity with respect to the number of jobs and machines when embedding disjunctive graphs. To further speed up solution evaluation, especially for batch processing, we design a novel message-passing mechanism that can evaluate multiple solutions simultaneously.

We verify our method on seven classic benchmarks. Extensive results show that our method generally outperforms state-of-the-art DRL-based methods by a large margin while maintaining a low computational cost. On large-scale instances, our method even outperforms Or-tools CP-SAT, a highly efficient constraint programming solver. Our aim is to showcase the effectiveness of Deep Reinforcement Learning (DRL) in learning superior search control policies compared to traditional methods. Our computationally efficient DRL-based heuristic narrows the performance gap with existing techniques and outperforms a tabu search algorithm in experiments. Furthermore, our method can potentially be combined with more complicated improvement frameworks, however it is out of the scope of this paper and will be investigated in the future.

## 2 RELATED WORK

Most existing DRL-based methods for JSSP focus on learning dispatching rules, or construction heuristics. Among them, L2D (Zhang et al., 2020) encodes partial solutions as disjunctive graphs and a GNN-based agent is trained via DRL to dispatch jobs to machines to construct tight schedules. Despite the superiority to traditional dispatching rules, its graph representation can only capture relations among dispatched operations, resulting in relatively large optimality gaps. A similar approach is proposed in (Park et al., 2021b). By incorporating heterogeneity of neighbours (e.g. predecessor or successor) in disjunctive graphs, a GNN model extracts separate embeddings and then aggregates them accordingly. While delivering better solutions than L2D, this method suffers from static graph representation, as the operations in each machine clique are always fully connected. Therefore, it fails to capture structural differences among solutions, which we notice to be critical. For example, given a JSSP instance, the processing order of jobs on each machine could be distinct for each solution, which cannot be reflected correctly in the respective representation scheme. ScheduleNet (Park et al., 2021a) introduces artificial machine nodes and edges in the disjunctive graph to encode machine-job relations. The new graph enriches the expressiveness but is still static. A Markov decision process formulation with a matrix state representation is proposed in (Tassel et al., 2021). Though the performance is ameliorated, it is an online method that learns for each instance separately, hence requiring much longer solving time. Moreover, unlike the aforementioned graph-based ones, the matrix representation in this method is not size-agnostic, which cannot be generalised

across different problem sizes. RASCL (Iklassov et al., 2022) learns generalized dispatching rules for solving JSSP, where the solving process is modelled as MDPs. Tassel et al. (2023a) propose to leverage existing Constraint Programming (CP) solvers to train a DRL-agent learning a Priority Dispatching Rule (PDR) that generalizes well to large instances. Despite substantial improvement, the performance of these works is still far from optimality compared with our method. Other learning-based construction heuristics for solving variants of JSSP include FFSP (Choo et al., 2022; Kwon et al., 2021) and scheduling problems in semiconductor manufacturing (Tassel et al., 2023b). However, these methods are incompatible with JSSP due to their underlying assumption of independent machine sets for each processing stage, which contradicts the shared machine configuration inherent in JSSP.

A work similar to ours is MGRO (Ni et al., 2021), which learns a policy to aid Iterated Greedy (IG), an improvement heuristic, to solve the hybrid flow shop scheduling problem. Our method differs from it in two aspects. Firstly, unlike MGRO which learns to pick local operators from a pool of manually designed ones, our policy does not require such manual work and directly outputs local moves. Secondly, MGRO encodes the current solution as multiple independent subgraphs, which is not applicable to JSSP since it cannot model precedence constraints.

## 3 PRELIMINARIES

**Job-shop Scheduling.** A JSSP instance of size $|\mathcal{J}| \times |\mathcal{M}|$ consists of $|\mathcal{J}|$ jobs and $|\mathcal{M}|$ machines. Each job $j \in \mathcal{J}$ is required to be processed by each machine $m \in \mathcal{M}$ in a predefined order $O_{j1}^{m_{j1}} \to \cdots \to O_{ji}^{m_{ji}} \to \cdots \to O_{j|\mathcal{M}|}^{m_{j|\mathcal{M}|}}$ with $O_{ji}^{m_{ji}}$ denoting the $i$th operation of job $j$, which should be processed by the machine $m_{ji}$ with processing time $p_{ji} \in \mathbb{N}$. Let $\mathcal{O}_j$ be the collections of all operations for job $j$. Only can the operation $O_{ji}^{m_{ji}}$ be processed when all its precedent operations $\{O_{jz}^{m_{jz}} | z < i\} \subset \mathcal{O}_j$ are processed, which is the so-called precedence constraint. The objective is to find a schedule $\eta : \{O_{ji}^{m_{ji}} | \forall j, i\} \to \mathbb{N}$, i.e., the starting time for each operation, such that the makespan $C_{\max} = \max(\eta(O_{ji}^{m_{ji}}) + p_{ji})$ is minimum without violating the precedence constraints. To enhance succinctness, the notation $m_{ji}$ indicating the machine allocation for operation $O_{ji}$ will be disregarded as machine assignments for operations remain constant throughout any given JSSP instance. This notation will be reinstated only if its specification is critical.

**Disjunctive Graph.** The disjunctive graph (Balas, 1969) is a directed graph $G = (\mathcal{O}, \mathcal{C} \cup \mathcal{D})$, where $\mathcal{O} = \{O_{ji} | \forall j, i\} \cup \{O_S, O_T\}$ is the set of all operations, with $O_S$ and $O_T$ being the dummy ones denoting the beginning and end of processing. $\mathcal{C}$ consists of directed arcs (conjunctions) representing the precedence constraints connecting every two consecutive operations of the same job. Undirected arcs (disjunctions) in set $\mathcal{D}$ connect operations requiring the same machine, forming a clique for each machine. Each arc is labelled with a weight, which is the processing time of the operation that the arc points from (two opposite-directed arcs can represent a disjunctive arc). Consequently, finding a solution $s$ to a JSSP instance is equivalent to fixing the direction of each disjunction, such that the resulting graph $G(s)$ is a DAG (Balas, 1969). The longest path from $O_S$ to $O_T$ in a solution is called the critical path (CP), whose length is the makespan of the solution. An example of disjunctive graph for a JSSP instance and its solution are depicted in Figure 1.

**The $N_5$ neighbourhood Structure.** Various neighbourhood structures have been proposed for JSSP (Zhang et al., 2007). Here we employ the well-known $N_5$ designed based on the concept of critical block (CB), i.e. a group of consecutive operations processed by the same machine on the critical path (refer to Figure 1 for an example). Given a solution $s$, $N_5$ constructs the neighbourhood $N_5(s)$ as follows. First, it finds the critical path CP(s) of $G(s)$ (randomly selects one if more than one exist). Then for each $CB_m = O_1^m \to O_2^m \to \cdots \to O_l^m \to \cdots \to O_{L-1}^m \to O_L^m$ with $m_l$ and $1 \le l \le L$ denoting the processing machine and the index of operation $O_l$ along the critical path CP(s), at

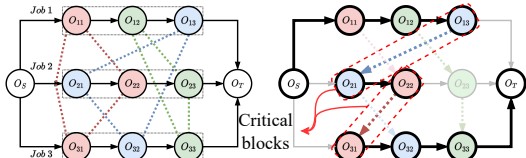

Figure 1: **Disjunctive graph representation.** Left: a $3 \times 3$ JSSP instance. Black arrows are conjunctions. Dotted lines of different colors are disjunctions, grouping operations on each machine into machine cliques. Right: a complete solution, where a critical path and a critical block are highlighted. Arc weights are omitted for clarity.

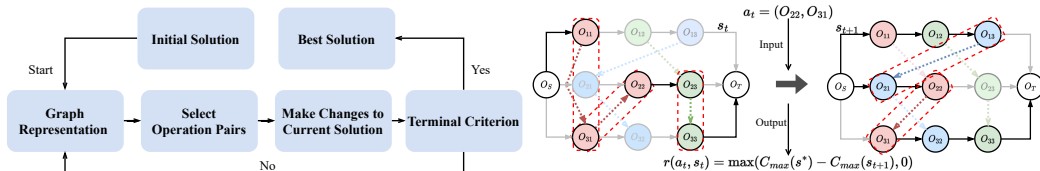

(a) The overall framework of our method.     (b) State transition example.

Figure 2: **Our local search framework and an example of state transition.** In Figure (b), the state $s_t$ is transited to $s_{t+1}$ by swapping operation $O_{22}$ and $O_{31}$. A new critical path with its two critical blocks is generated and highlighted in $s_{t+1}$.

most two neighbours are obtained by swapping the first $(O_1^{m_1}, O_2^{m_2})$ or last pair $(O_{L-1}^{m_L-1}, O_L^{m_L})$ of operations. Only one neighbour exists if a CB has only two operations. For the first and last CB, only the last and first operation pair are swapped. Consequently, the neighbourhood size $|N_5(s)|$ is at most $2N(s) - 2$ with $N(s)$ being the number of CBs in $G(s)$.

## 4 METHODOLOGY

The overall framework of our improvement method is shown in Figure 2(a). The initial solution is generated by basic dispatching rules. During the improvement process, solutions are proxied by their graphical counterpart, i.e. disjunctive graphs. Different from traditional improvement heuristics which need to evaluate all neighbours at each step, our method directly outputs an operation pair, which is used to update the current solution according to $N_5$. The process terminates if certain condition is met. Here we set it as reaching a searching horizon of $T$ steps.

Below we present our DRL-based method for learning the pair picking policy. We first formulate the learning task as a Markov decision process (MDP). Then we show how to parameterize the policy based on GNN, and design a customized REINFORCE (Williams, 1992) algorithm to train the policy network. Finally, we design a message-passing mechanism to quickly calculate the schedule, which significantly improves efficiency especially for batch training. Note that, the proposed policy possesses linear computational complexity w.r.t. the number of jobs $|\mathcal{J}|$ and machines $|\mathcal{M}|$.

### 4.1 THE SEARCHING PROCEDURE AS AN MDP

**States.** A state $s_t \in \mathcal{S}$ is the complete solution at step $t$, with $s_0$ being the initial one. Each state is represented as its disjunctive graph. Features of each node $O_{ji}$ is collected in a vector $\mathbf{x}_{ji} = (p_{ji}, est_{ji}, lst_{ji}) \in \mathbb{R}^3$, where $est_{ji}$ and $lst_{ji}$ are the earliest and latest starting time of $O_{ji}$, respectively. For a complete solution, $est(O_{ji})$ is the actual start time of $O_{ji}$ in the schedule. If $O_{ji}$ is located on a critical path, then $est_{ji} = lst_{ji}$ (Jungnickel & Jungnickel, 2005). This should be able to help the agent distinguish whether a node is on the critical path.

**Actions.** Since we aim at selecting a solution within the neighbourhood, an action $a_t$ is one of the operation pairs $(O_{ji}, O_{kl})$ in the set of all candidate pairs defined by $N_5$. Note that the action space $A_t$ is dynamic w.r.t each state.

**Transition.** The next state $s_{t+1}$ is derived deterministically from $s_t$ by swapping the two operations of action $a_t = (O_{ji}, O_{kl})$ in the graph. An example is illustrated in Figure 2(b), where features $est$ and $lst$ are recalculated for each node in $s_{t+1}$. If there is no feasible action $a_t$ for some $t$, e.g. there is only one critical block in $s_t$, then the episode enters an *absorbing* state and stays within it. We present a example in Appendix J to facilitate better understanding the state transition mechanism.

**Rewards.** Our ultimate goal is to improve the initial solution as much as possible. To this end, we design the reward function as follows:

$$r(s_t, a_t) = \max\left(C_{max}(s_t^*) - C_{max}(s_{t+1}), 0\right),$$

(1)

where $s_t^*$ is the best solution found until step $t$ (the incumbent), which is updated only if $s_{t+1}$ is better, i.e. $C_{max}(s_{t+1}) < C_{max}(s_t^*)$. Initially, $s_0^* = s_0$. The cumulative reward until $t$ is $R_t = \sum_{t'=0}^{t} r(s_{t'}, a_{t'}) = C_{max}(s_0) - C_{max}(s_t^*)$, which is exactly the improvement against initial

Figure 3: **The architecture of our policy network.**

solution $s_0$. When the episode enters the absorbing state, by definition, the reward is always 0 since there is no possibility of improving the incumbent from then on.

**Policy.** For state $s_t$, a stochastic policy $\pi(a_t|s_t)$ outputs a distribution over the actions in $A_t$. If the episode enters the absorbing state, the policy will output a dummy action which has no effect.

## 4.2 POLICY MODULE

We parameterize the stochastic policy $\pi_\theta(a_t|s_t)$ as a GNN-based architecture with parameter set $\theta$. A GNN maps the graph to a continuous vector space. The embeddings of nodes could be deemed as feature vectors containing sufficient information to be readily used for various downstream tasks, such as selecting operation pairs for local moves in our case. Moreover, a GNN-based policy is able to generalize to instances of various sizes that are unseen during training, due to its size-agnostic property. The architecture of our policy network is shown in Figure 3.

### 4.2.1 GRAPH EMBEDDING

For the disjunctive graph, we define a node $U$ as a neighbour of node $V$ if an arc points from $U$ to $V$. Therefore, the dummy operation $O_S$ has no neighbours (since there are no nodes pointing to it), and $O_T$ has $|\mathcal{J}|$ neighbours (since every job's last operation is pointing to it). There are two notable characteristics of this graph. First, the topology is dynamically changing due to the MDP state transitions. Such topological difference provides rich information for distinguishing two solutions from the structural perspective. Second, most operations in the graph have two different types of neighbours, i.e. its job predecessor and machine predecessor. These two types of neighbours hold respective semantics. The former is closely related to the precedence constraints, while the latter reflects the processing sequence of jobs on each machine. Based on this observation, we propose a novel GNN with two independent modules to embed disjunctive graphs effectively. We will experimentally prove by an ablation study (Appendix M) that the combination of the two modules are indeed more effective.

**Topological Embedding Module.** For two different solutions, there must exist a machine on which jobs are processed in different orders, i.e. the disjunctive graphs are topologically different. The first module, which we call *topological embedding module* (TPM), is expected to capture such structural differences, so as to help the policy network distinguish different states. To this end, we exploit Graph Isomorphism Network (GIN) (Xu et al., 2019), a well-known GNN variant with strong discriminative power for non-isomorphic graphs as the basis of TPM. Given a disjunctive graph $G$, TPM performs $K$ iterations of updates to compute a $p$-dimensional embeddings for each node $V \in \mathcal{O}$, and the update at iteration $k$ is expressed as follows:

$$\mu_V^k = \text{MLP}_T^k((1 + \epsilon^k)\mu_V^{k-1} + \sum_{U \in \mathcal{N}(V)} \mu_U^{k-1}), \tag{2}$$

where $\mu_V^k$ is the topological embedding of node $V$ at iteration $k$ and $\mu_V^0 = \mathbf{x}_{ji}$ is its raw features, $\text{MLP}_T^k$ is a Multi-Layer Perceptron (MLP) for iteration $k$, $\epsilon^k$ is an arbitrary number that can be learned, and $\mathcal{N}(V)$ is the neighbourhood of $V$ in $G$. For each $k$, we attain a graph-level embedding $\mu_G^k \in \mathbb{R}^p$ by an average pooling function $L$ with embeddings of all nodes as input as follows:

$$\mu_G^k = L(\{\mu_V^k : V \in \mathcal{O}\}) = \frac{1}{|\mathcal{O}|} \sum_{V \in \mathcal{O}} \mu_V^k. \tag{3}$$

Finally, the topological embeddings of each node and the disjunctive graph after the $K$ iterations are $\{\mu_V = \sum_k \mu_V^k : \forall V \in \mathcal{O}\}$ and $\mu_G = \sum_k \mu_G^k$, respectively. For representation learning with GIN, layers of different depths learn different structural information (Xu et al., 2019). The representations acquired by the shallow layers sometimes generalize better, while the deeper layers are more oriented for the target task. To consider all the discrepant structural information of the disjunctive graph and align the hidden dimension, we sum the embeddings from all layers.

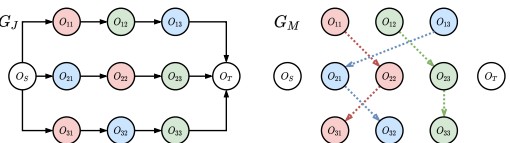

Figure 4: **Example of** $G_J$ **and** $G_M$.

**Context-aware Embedding Module.** Now we present the second module to capture information from the two types of neighbouring nodes, which we call *context-aware embedding module* (CAM). Specifically, we separate a disjunctive graph $G$ into two subgraphs $G_J$ and $G_M$, i.e. contexts, as shown in Figure 4. Both subgraphs contain all nodes of $G$, but have different sets of edges to reflect respective information. $G_J$ contains only conjunctive arcs which represent precedence constraints, while $G_M$ contains only (directed) disjunctive arcs to represent the job processing order on each machine. Note that the two dummy nodes $O_S$ and $O_T$ are isolated in $G_M$, since they are not involved in any machine.

Similar to TPM, CAM also updates the embeddings for $K$ iterations by explicitly considering the two contexts during message passing and aggregation. Particularly, the update of node embeddings at CAM iteration $k$ is given as follows:

$$\nu_V^{J,k} = \text{GAT}_J^k \left( \nu_V^{k-1}, \{\nu_U^{k-1} | U \in \mathcal{N}_J(V)\} \right), \tag{4}$$

$$\nu_V^{M,k} = \text{GAT}_M^k \left( \nu_V^{k-1}, \{\nu_U^{k-1} | U \in \mathcal{N}_M(V)\} \right), \tag{5}$$

$$\nu_V^k = \frac{1}{2} \left( \nu_V^{J,k} + \nu_V^{M,k} \right), \tag{6}$$

where $\text{GAT}_J^k$ and $\text{GAT}_M^k$ are two graph attention layers (Veličković et al., 2018) with $n_h$ attention heads, $\nu_V^k \in \mathbb{R}^p$ is the context-aware embedding for $V$ at layer $k$, $\nu_V^{J,k} \in \mathbb{R}^p$ and $\nu_V^{M,k} \in \mathbb{R}^p$ are $V$'s embedding for precedence constraints and job processing sequence, and $\mathcal{N}_J(V)$ and $\mathcal{N}_M(V)$ are $V$'s neighbourhood in $G_J$ and $G_M$, respectively. For the first iteration, we initialize $\nu_V^0 = \mathbf{x}_{ji}, \forall V \in \mathcal{O}$. Finally, we compute the graph-level context-aware embedding $\nu_G$ using average pooling as follows:

$$\nu_G = \frac{1}{|\mathcal{O}|} \sum_{V \in \mathcal{O}} \nu_V^K. \tag{7}$$

To generate a global representation for the whole graph, we merge the topological and context-aware embeddings by concatenating them and yield:

$$\{h_V = (\mu_V^K : \nu_V^K) | \forall V \in \mathcal{O}\}, h_G = (\mu_G : \nu_G). \tag{8}$$

***Remark.*** We choose GIN as the building block for TPM mainly because it is one of the strongest GNN architectures with proven power to discriminate graphs from a topological point of view. It may benefit identifying topological discrepancies between solutions. As for GAT, it is an equivalent counterpart of Transformers (Vaswani et al., 2017) for graphs, which is a dominant architecture for learning representations from element attributes. Therefore we adopt GAT as the building block of our CAM module to extract node embeddings for the heterogeneous contexts.

### 4.2.2 ACTION SELECTION

Given the node embeddings $\{h_V\}$ and graph embedding $h_G$, we compute the "score" of selecting each operation pair as follows. First, we concatenate $h_G$ to each $h_V$, and feed it into a network $\text{MLP}_A$, to obtain a latent vector denoted as $h'_V$ with dimension $q$, which is collected as a matrix $h'$ with dimension $|\mathcal{O}| \times q$. Then we multiply $h'$ with its transpose $h'^\mathsf{T}$ to get a matrix $SC$ with dimension $|\mathcal{O}| \times |\mathcal{O}|$ as the score of picking the corresponding operation pair. Next, for all pairs that are not included in the current action space, we mask them by setting their scores to $-\infty$. Finally,

we flatten the score matrix and apply a softmax function to obtain the probability of selecting each feasible action.

We present the theoretical computational complexity of the proposed policy network. Specifically, for a JSSP instance of size $|\mathcal{J}| \times |\mathcal{M}|$, we can show that:

**Theorem 4.1.** *The proposed policy network has linear time complexity w.r.t both $|\mathcal{J}|$ and $|\mathcal{M}|$.*

The detailed proof is presented in Appendix C. In the experiments, we will also provide empirical analysis and comparison with other baselines.

### 4.3 THE $n$-STEP REINFORCE ALGORITHM

We propose an $n$-step REINFORCE algorithm for training the policy network. The motivation, benefits, and details of this algorithm are presented in Appendix A.

### 4.4 MESSAGE-PASSING FOR CALCULATING SCHEDULE

The improvement process requires evaluating the quality of neighbouring solutions by calculating their schedules. In traditional scheduling methods, this is usually done by using the critical path method (CPM) (Jungnickel & Jungnickel, 2005), which calculates the start time for each node recursively. However, it can only process a single graph and cannot make full use of the parallel computing power of GPU. Therefore, traditional CPM is time-costly in processing a batch of instances. To alleviate this issue, we propose an equivalent variant of this algorithm using a message-passing mechanism motivated by the computation of GNN, which enables batch processing and is compatible with GPU.

Our message-passing mechanism works as follows. Given a directed disjunctive graph $G$ representing a solution $s$, we maintain a message $ms_V = (d_V, c_V)$ for each node $V \in \mathcal{O}$, with initial values $d_V = 0$ and $c_V = 1$, except $c_V = 0$ for $V = O_S$. Let $mp_{max}(\cdot)$ denote a message-passing operator with max-pooling as the neighbourhood aggregation function, based on which we perform a series of message updates. During each of them, for $V \in \mathcal{O}$, we update its message by $d_V = mp_{max}(\{p_U + (1 - c_U) \cdot d_U | \forall U \in \mathcal{N}(V)\})$ and $c_V = mp_{max}(\{c_U | \forall U \in \mathcal{N}(V)\})$ with $p_V$ and $\mathcal{N}(V)$ being the processing time and the neighbourhood of $V$, respectively. Let $H$ be the number of nodes on the path from $O_S$ to $O_T$ containing the most nodes. Then we can show that:

**Theorem 4.2.** *After at most $H$ times of message passing, $d_V = est_V, \forall V \in \mathcal{O}$, and $d_T = C_{max}(s)$.*

The proof is presented in Appendix D.1. This proposition indicates that our message-passing evaluator is equivalent to CPM. It is easy to prove that this proposition also holds for a batch of graphs. Thus the practical run time for calculating the schedule using our evaluator is significantly reduced due to computation in parallel across the batch. We empirically verify the effectiveness by comparing it with CPM (Appendix D.3).

Similarly, $lst_V$ can be calculated by a backward version of our message-passing evaluator where the message is passed from node $O_T$ to $O_S$ in a graph $\overline{G}$ with all edges reversed. Each node $V$ maintains a message $\overline{ms}_V = (\bar{d}_V, \bar{c}_V)$. The initial values are $\bar{d}_V = -1$ and $\bar{c}_V = 1$, except $\bar{d}_T = -C_{max}(s)$ and $\bar{c}_T = 0$ for $V = O_T$. The message of $V$ is updated as $\bar{d}_V = mp_{max}(\{p_U + (1 - \bar{c}_U) \cdot \bar{d}_U | \forall U \in \mathcal{N}(V)\})$ and $\bar{c}_V = mp_{max}(\{\bar{c}_U | \forall U \in \mathcal{N}(V)\})$. We can show that:

**Corollary 4.3.** *After at most $H$ times of message passing, $\bar{d}_V = -lst_V, \forall V \in \mathcal{O}$, and $\bar{d}_S = 0$.*

The proof is presented in Appendix D.2. Please refer to an example in D.4 for the procedure of computing $est$ using the proposed message-passing operator.

## 5 EXPERIMENTS

### 5.1 EXPERIMENTAL SETUP

**Datasets.** We perform training on synthetic instances generated following the widely used convention in (Taillard, 1993). We consider five problem sizes, including $10\times10$, $15\times10$, $15\times15$, $20\times10$, and $20\times15$. For evaluation, we perform testing on seven widely used classic benchmarks unseen

Table 1: **Performance on classic benchmarks.** "Gap": the average gap to the best solutions in the literature. "Time": the average time of solving a single instance ("s", "m", and "h" means seconds, minutes, and hours, respectively.). For each problem size, results in bold and bold blue represent the local and overall best results, respectively.

| Method | Taillard 15×15 | | 20×15 | | 20×20 | | 30×15 | | 30×20 | | 50×15 | | 50×20 | | 100×20 | | ABZ 10×10 | | 20×15 | | FT 6×6 | | 10×10 | | 20×5 | |
|---|---|---|---|---|---|---|---|---|---|---|---|---|---|---|---|---|---|---|---|---|---|---|---|---|---|---|
| | Gap | Time | Gap | Time | Gap | Time | Gap | Time | Gap | Time | Gap | Time | Gap | Time | Gap | Time | Gap | Time | Gap | Time | Gap | Time | Gap | Time | Gap | Time |
| CP-SAT | 0.1% | 7.7m | 0.2% | 0.8h | 0.7% | 1.0h | 2.1% | 1.0h | 2.8% | 1.0h | **0.0%** | 0.4h | 2.8% | 0.9h | 3.9% | 1.0h | **0.0%** | 0.8s | 1.0% | 1.0h | **0.0%** | 0.1s | **0.0%** | 4.1s | **0.0%** | 4.8s |
| L2D | 24.7% | 0.4s | 30.0% | 0.6s | 28.8% | 1.1s | 30.5% | 1.3s | 32.9% | 1.5s | 20.0% | 2.2s | 23.9% | 3.6s | 12.9% | 28.2s | 14.8% | 0.1s | 24.9% | 0.6s | 14.5% | 0.1s | 21.0% | 0.2s | 36.3% | 0.2s |
| RL-GNN | 20.1% | 3.0s | 24.9% | 7.0s | 29.2% | 12.0s | 24.7% | 24.7s | 32.0% | 38.0s | 15.9% | 1.9m | 21.3% | 3.2m | 9.2% | 28.2m | 10.1% | 0.5s | 29.0% | 7.3s | 29.1% | 0.1s | 22.8% | 0.5s | 14.8% | 1.3s |
| ScheduleNet | 15.3% | 3.5s | 19.4% | 6.6s | 17.2% | 11.0s | 19.1% | 17.1s | 23.7% | 28.3s | 13.9% | 52.5s | 13.5% | 1.6m | **6.7%** | 7.4m | 6.1% | 0.7s | 20.5% | 6.6s | 7.3% | 0.2s | 19.5% | 0.8s | 28.6% | 1.6s |
| GD-500 | 11.9% | 48.2s | 14.4% | 75.2s | 15.7% | 1.7m | 17.9% | 91.3s | 20.1% | 1.7m | 12.5% | 2.1m | 13.7% | 2.6m | 7.3% | 4.6m | 6.2% | 26.8s | 16.5% | 58.8s | 3.6% | 15.8s | 10.1% | 33.2s | 9.8% | 37.0s |
| FI-500 | 12.3% | 70.6s | 15.7% | 87.4s | 14.5% | 2.2m | 18.4% | 1.9m | 22.0% | 3.0m | 14.2% | 2.5m | 15.6% | 4.3m | 9.3% | 7.4m | 3.5% | 33.8s | 16.7% | 85.4s | **0.0%** | 18.2s | 10.1% | 42.4s | 16.1% | 40.7s |
| BI-500 | 11.7% | 65.4s | 14.5% | 83.5s | 14.3% | 2.3m | 18.3% | 1.9m | 20.7% | 2.9m | 13.1% | 2.7m | 14.2% | 4.0m | 8.1% | 7.0m | 4.1% | 31.2s | 16.8% | 84.9s | **0.0%** | 18.0s | 12.9% | 41.3s | 22.0% | 40.6s |
| Ours-500 | **9.3%** | 9.3s | **11.6%** | 10.1s | **12.4%** | 10.9s | **14.7%** | 12.7s | **17.5%** | 14.0s | **11.0%** | 16.2s | **13.0%** | 22.8s | 7.9% | 50.2s | **2.8%** | 7.4s | **11.9%** | 10.2s | **0.0%** | 6.8s | **9.9%** | 7.5s | **6.1%** | 7.4s |
| GD-5000 | 11.9% | 7.7m | 14.4% | 12.3m | 15.7% | 17.3m | 17.9% | 15.3m | 20.1% | 16.6m | 12.5% | 20.0m | 13.7% | 23.1m | 7.3% | 39.2m | 6.2% | 4.5m | 16.5% | 9.7m | 3.6% | 2.6m | 10.1% | 5.5m | 9.8% | 6.1m |
| FI-5000 | 9.8% | 12.6m | 13.0% | 15.8m | 13.3% | 24.2m | 15.0% | 20.2m | 17.5% | 30.6m | 10.5% | 27.4m | 11.8% | 44.5m | 6.4% | 76.2m | 2.7% | 5.9m | 13.3% | 15.3m | **0.0%** | 3.0m | **5.6%** | 7.3m | 6.9% | 6.9m |
| BI-5000 | 10.5% | 11.9m | 11.8% | 15.6m | 12.0% | 23.6m | 14.4% | 19.6m | 16.9% | 28.8m | 9.2% | 27.5m | 10.9% | 43.4m | 5.4% | 76.4m | 2.1% | 5.7m | 10.9% | 15.0m | **0.0%** | 3.1m | 6.2% | 7.1m | 6.6% | 7.0m |
| Ours-1000 | 8.6% | 18.7s | 10.4% | 20.3s | 11.4% | 22.2s | 12.9% | 24.7s | 15.7% | 28.4s | 9.0% | 32.9s | 11.4% | 45.4s | 6.6% | 1.7m | 2.8% | 15.0s | 11.2% | 19.9s | **0.0%** | 13.5s | 8.0% | 15.1s | 3.9% | 15.0s |
| Ours-2000 | 7.1% | 37.7s | 9.4% | 41.5s | 10.2% | 44.7s | 11.0% | 49.1s | 14.0% | 56.8s | 6.9% | 65.7s | 9.3% | 90.9s | 5.1% | 3.4m | 2.8% | 30.1s | 9.5% | 39.3s | **0.0%** | 27.2s | 5.7% | 30.0s | 1.5% | 29.9s |
| Ours-5000 | **6.2%** | 92.2s | **8.3%** | 1.7m | **9.0%** | 1.9m | **9.0%** | 2.0m | **12.6%** | 2.4m | **4.6%** | 2.8m | **6.5%** | 3.8m | **3.0%** | 8.4m | **1.4%** | 75.2s | **8.6%** | 99.6s | **0.0%** | 67.7s | **5.6%** | 74.8s | **1.1%** | 73.3s |

| Method | LA 10×5 | | 15×5 | | 20×5 | | 10×10 | | 15×10 | | 20×10 | | 30×10 | | 15×15 | | SWV 20×10 | | 20×15 | | 50×10 | | ORB 10×10 | | YN 20×20 | |
|---|---|---|---|---|---|---|---|---|---|---|---|---|---|---|---|---|---|---|---|---|---|---|---|---|---|---|
| | Gap | Time | Gap | Time | Gap | Time | Gap | Time | Gap | Time | Gap | Time | Gap | Time | Gap | Time | Gap | Time | Gap | Time | Gap | Time | Gap | Time | Gap | Time |
| CP-SAT | 0.0% | 0.1s | 0.0% | 0.2s | 0.0% | 0.5s | 0.0% | 0.4s | 0.0% | 21.0s | 0.0% | 12.9m | 0.0% | 13.7s | 0.0% | 30.1s | 0.1% | 0.8h | 2.5% | 1.0h | 1.6% | 0.5h | 0.0% | 4.8s | 0.5% | 1.0h |
| L2D | 14.3% | 0.1s | 5.5% | 0.1s | 4.2% | 0.2s | 21.9% | 0.1s | 24.6% | 0.2s | 24.7% | 0.4s | 8.4% | 0.7s | 27.1% | 0.4s | 41.4% | 0.3s | 40.6% | 0.6s | 30.8% | 1.2s | 31.8% | 0.1s | 22.1% | 0.9s |
| RL-GNN | 16.1% | 0.2s | 1.1% | 0.5s | 2.1% | 1.2s | 17.1% | 0.5s | 22.0% | 1.5s | 27.3% | 3.3s | 6.3% | 11.3s | 21.4% | 2.8s | 28.4% | 3.4s | 29.4% | 7.2s | **16.8%** | 51.5s | 21.8% | 0.5s | 24.8% | 11.0s |
| ScheduleNet | 12.1% | 0.6s | 2.7% | 1.2s | 3.6% | 1.9s | 11.9% | 0.8s | 14.6% | 2.0s | 15.7% | 4.1s | 3.1% | 9.3s | 16.1% | 3.5s | 34.4% | 3.9s | 30.5% | 6.7s | 25.3% | 25.1s | 20.0% | 0.8s | 18.4% | 11.2s |
| GD-500 | 4.8% | 16.1s | 0.6% | 23.2s | 0.3% | 31.4s | 5.8% | 26.5s | 10.4% | 39.3s | 11.2% | 46.9s | 2.4% | 58.8s | 9.5% | 49.7s | 33.7% | 61.7s | 29.1% | 78.3s | 22.0% | 2.1m | 11.6% | 36.6s | 14.5% | 86.3s |
| FI-500 | 4.5% | 17.5s | 0.1% | 22.8s | 0.5% | 30.9s | 5.9% | 35.3s | 8.4% | 49.9s | 13.7% | 59.0s | 2.9% | 74.9s | 10.3% | 63.4s | 32.3% | 75.3s | 31.0% | 1.7m | 23.3% | 2.4m | 10.7% | 46.0s | 18.8% | 2.4m |
| BI-500 | 4.2% | 19.8s | **0.0%** | 22.8s | 0.5% | 30.9s | 5.1% | 34.3s | 8.9% | 47.3s | 10.9% | 60.5s | 2.7% | 73.1s | 10.1% | 62.5s | 33.5% | 75.2s | 29.7% | 1.7m | 22.2% | 2.5m | 11.3% | 42.5s | 15.1% | 2.1m |
| Ours-500 | 2.1% | 6.9s | **0.0%** | 6.8s | **0.0%** | 7.1s | **4.4%** | 7.5s | **6.4%** | 8.0s | **7.0%** | 8.9s | **0.2%** | 10.2s | **7.3%** | 9.0s | **29.6%** | 8.8s | **25.5%** | 9.7s | 21.4% | 12.5s | **8.2%** | 7.4s | **12.4%** | 11.7s |
| GD-5000 | 4.8% | 2.7m | 0.6% | 3.9m | 0.3% | 5.2m | 5.8% | 4.4m | 10.4% | 6.5m | 11.2% | 7.8m | 2.4% | 9.7m | 9.5% | 8.3m | 33.7% | 10.2m | 29.1% | 13.0m | 22.0% | 20.7m | 11.6% | 6.1m | 14.5% | 14.1m |
| FI-5000 | 2.9% | 2.8m | **0.0%** | 3.8m | **0.0%** | 5.1m | 3.6% | 6.2m | 6.1% | 8.3m | 8.3% | 9.8m | 0.3% | 9.8m | 8.4% | 12.0m | 25.9% | 11.9m | 25.8% | 18.0m | 21.3% | 24.4m | 8.2% | 7.7m | 13.7% | 24.9m |
| BI-5000 | 1.9% | 2.8m | **0.0%** | 3.8m | 0.1% | 5.1m | 5.0% | 6.2m | 6.0% | 8.6m | 6.1% | 10.1m | 0.2% | 9.6m | 9.0% | 11.8m | 25.5% | 11.8m | 25.2% | 17.8m | 20.6% | 24.6m | 8.0% | 7.7m | 12.0% | 22.1m |
| Ours-1000 | **1.8%** | 14.0s | **0.0%** | 13.9s | **0.0%** | 14.5s | 2.3% | 15.0s | 5.1% | 16.0s | 5.7% | 17.5s | **0.0%** | 20.4s | 6.6% | 18.2s | 24.5% | 17.6s | 23.5% | 19.0s | 20.1% | 25.4s | 6.6% | 15.0s | 10.5% | 23.4s |
| Ours-2000 | **1.8%** | 27.9s | **0.0%** | 28.3s | **0.0%** | 28.7s | 1.8% | 30.1s | 4.0% | 32.2s | 3.4% | 34.2s | **0.0%** | 40.4s | 6.3% | 35.9s | 21.8% | 34.7s | 21.7% | 38.8s | 19.0% | 49.5s | 5.7% | 29.9s | 9.6% | 47.0s |
| Ours-5000 | **1.8%** | 70.0s | **0.0%** | 71.0s | **0.0%** | 73.7s | **0.9%** | 75.1s | **3.4%** | 80.9s | **2.6%** | 85.4s | **0.0%** | 99.3s | **5.9%** | 88.8s | **17.8%** | 86.9s | **17.0%** | 99.8s | **17.1%** | 2.1m | **3.8%** | 75.9s | **8.7%** | 1.9m |

in training[2], including Taillard (Taillard, 1993), ABZ (Adams et al., 1988), FT (Fisher, 1963), LA (Lawrence, 1984), SWV (Storer et al., 1992), ORB (Applegate & Cook, 1991), and YN (Yamada & Nakano, 1992). Since the upper bound found in these datasets is usually obtained with different SOTA metaheuristic methods (e.g., (Constantino & Segura, 2022)), we have implicitly compared with them although we did not explicitly list those metaheuristic methods. It is also worth noting that the training instances are generated by following distributions different from these benchmarking datasets. Therefore, we have also implicitly tested the zero-shot generalization performance of our method. Moreover, we consider three extremely large datasets (up to 1000 jobs), where our method outperforms CP-SAT by a large margin. The detailed results are presented in Appendix L.

**Model and configuration.** Please refer to Appendix B for the hardware and training (and testing) configurations of our method. Our code and data are publicly available at https://github.com/zcaicaros/L2S.

**Baselines.** We compare our method with three state-of-the-art DRL-based methods, namely L2D (Zhang et al., 2020), RL-GNN (Park et al., 2021b), and ScheduleNet (Park et al., 2021a). The online DRL method in (Tassel et al., 2021) is only compared on Taillard 30×20 instances for which they report results. We also compare with three hand-crafted rules widely used in improvement heuristics, i.e. greedy (GD), best-improvement (BI) and first-improvement (FI), to verify the effectiveness of automatically learning improvement policy. For a fair comparison, we let them start from the same initial solutions as ours, and allow BI and FI to restart so as to escape local minimum. Also, we equip them with the message-passing evaluator, which can significantly speed up their calculation since they need to evaluate the whole neighbourhood at each step. More details are presented in Appendix E. The last major baseline is the highly efficient constraint programming solver CP-SAT (Perron & Furnon, 2019) in Google OR-Tools, which has strong capability in solving scheduling problems (Da Col & Teppan, 2019), with 3600 seconds time limit. We also compare our method with an advanced tabu search algorithm (Zhang et al., 2007). The results demonstrate the advantageous efficiency of our approach in selecting moves and achieving better solutions within the same computational time (see Section H for details).

---

[2]The best-known results for these public benchmarks are available in http://optimizer.com/TA.php and http://jobshop.jjvh.nl/.

## 5.2 Performance on Classic Benchmarks

We first evaluate our method on the seven classic benchmarks, by running 500 improvement steps as in training. Here, we solve each instance using the model trained with the closest size. In Appendix I, we will show that the performance of our method can be further enhanced by simply assembling all the trained models. For the hand-crafted rules, we also run them for 500 improvement steps. We reproduce RL-GNN and ScheduleNet since their models and code are not publicly available. Note that to have a fair comparison of the run time, for all methods, we report the average time of solving a single instance without batching. Results are gathered in Table 1 (upper part in each of the two major rows). We can observe that our method is computationally efficient, and almost consistently outperforms the three DRL baselines with only 500 steps. RL-GNN and ScheduleNet are relatively slower than L2D, especially for large problem sizes, because they adopt an event-driven based MDP formulation. On most of the instances larger than $20 \times 15$, our method is even faster than the best learning-based construction heuristic ScheduleNet, and meanwhile delivers much smaller gaps. In Appendix C.2, we will provide a more detailed comparison on the efficiency with RL-GNN and ScheduleNet. With the same 500 steps, our method also consistently outperforms the conventional hand-crafted improvement rules. This shows that the learned policies are indeed better in terms of guiding the improvement process. Moreover, our method is much faster than the conventional ones, which verifies the advantage of our method in that the neighbour selection is directly attained by a neural network, rather than evaluating the whole neighbourhood.

## 5.3 Generalization to Larger Improvement Steps

We further evaluate the capability of our method in generalizing to larger improvement steps (up to 5000). Results are also gathered in Table 1 (**lower part** in each major row), where we report the results for hand-crafted rules after 5000 steps. We can observe that the improvement policies learned by our agent for 500 steps generalize fairly well to a larger number of steps. Although restart is allowed, the greedy rule (GD) stops improving after 500 steps due to the appearance of "cycling" as it could be trapped by repeatedly selecting among several moves (Nowicki & Smutnicki, 1996). This is a notorious issue that causes the failure to hand-craft rules for JSSP. However, our agent could automatically learn to escape the cycle even without restart, by maximizing the long-term return. In addition, it is worth noticing that our method with 5000 steps is the only one that exceeds CP-SAT on Taillard $100 \times 20$ instances, the largest ones in these benchmarks. Besides the results in Table 1, our method also outperforms the online DRL-based method (Tassel et al., 2021) on Taillard $30 \times 20$ instances, which reports a gap of 13.08% with 600s run time, while our method with 5000 steps produces a gap of 12.6% in 144s. The detailed results are presented in Appendix F. It is also apparent from Table 1 that the run time of our model is linear w.r.t the number of improvement steps $T$ for any problem size.

## 5.4 Comparison with tabu search

Due to the page limit, we refer readers to Appendix H for details. In a nutshell, the performance of our method is comparable to that of tabu search when using the same number of improvement steps but with a significant speed advantage. On the other hand, our method outperforms tabu search when the computational time (wall clock time) is the same.

## 6 Conclusions and Future Work

This paper presents a DRL-based method to learn high-quality improvement heuristics for solving JSSP. By leveraging the advantage of disjunctive graph in representing complete solutions, we propose a novel graph embedding scheme by fusing information from the graph topology and heterogeneity of neighbouring nodes. We also design a novel message-passing evaluator to speed up batch solution evaluation. Extensive experiments on classic benchmark instances well confirm the superiority of our method to the state-of-the-art DRL baselines and hand-crafted improvement rules. Our methods reduce the optimality gaps on seven classic benchmarks by a large margin while maintaining a reasonable computational cost. In the future, we plan to investigate the potential of applying our method to more powerful improvement frameworks to further boost their performance.

## 7 ACKNOWLEDGEMENT

This research was supported by the Singapore Ministry of Education (MOE) Academic Research Fund (AcRF) Tier 1 grant. This work is also partially supported by the MOE AcRF Tier 1 funding (RG13/23). This research was also supported by the National Natural Science Foundation of China under Grant 62102228, and the Natural Science Foundation of Shandong Province under Grant ZR2021QF063. The authors sincerely thank Yifan Yang from the School of Engineering and Computer Science of Victoria University of Wellington for her help on improving the camera-ready version of the manuscript.

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

## A  THE $n$-STEP REINFORCE ALGORITHM

We adopt the REINFORCE algorithm (Williams, 1992) for training. However, here the vanilla REINFORCE may bring undesired challenges in training for two reasons. First, the reward is sparse in our case, especially when the improvement process becomes longer. This is a notorious reason causing various DRL algorithms to fail (Nair et al., 2018; Riedmiller et al., 2018). Second, it will easily cause out-of-memory issue if we employ a large step limit $T$, which is often required for desirable improvement performance. To tackle these challenges, we design an $n$-step version of REINFORCE, which trains the policy every $n$ steps along the trajectory (the pseudo-code is given below). Since the $n$-step REINFORCE only requires storing every $n$ steps of transitions, the agent can explore a much longer episode. This training mechanism also helps deal with sparse reward, as the agent will be trained first on the data at the beginning where the reward is denser, such that it is ready for the harder part when the episode goes longer.

We present the pseudo-code of the $n$-step REINFORCE algorithm in Algorithm 1.

---

**Algorithm 1** $n$-step REINFORCE

**Input**: Policy $\pi_\theta(a_t|s_t)$, step limit $T$, step size $n$, learning rate $\alpha$, training problem size $|\mathcal{J}| \times |\mathcal{M}|$, batch size $B$, total number of training instances $I$
**Output**: Trained policy $\pi_{\theta*}(a_t|s_t)$

1: **for** $i = 0$ to $i < I$ **do**
2:    Randomly generate $B$ instances of size $|\mathcal{J}| \times |\mathcal{M}|$, and compute their initial solutions $\{s_0^1, ..., s_0^B\}$ by using the dispatching rule FDD/MWKR
3:    Initialize a training data buffer $D^b$ with size 0 for each $s_0^b \in \{s_0^1, ..., s_0^B\}$;
4:    **for** $t = 0$ to $T$ **do**
5:        **for** $s_t^b \in \{s_t^1, ..., s_t^B\}$ **do**
6:            Compute a local move $a_t^b \sim \pi_\theta(a_t^b|s_t^b)$
7:            Update $s_t^b$ w.r.t $a_t^b$ and receive a reward $r(a_t^b, s_t^b)$
8:            **if** $t \bmod n = 0$ **then**
9:                $loss_\theta^b = 0$
10:               **for** $j = n$ to $0$ **do**
11:                   $loss_\theta^b += -\log \pi_\theta(a_{t-j}^b|s_{t-j}^b) \cdot R_{t-j}^b$, where $R_{t-j}^b$ is the return for step $t-j$
12:               **end for**
13:               $\theta = \theta + \alpha \nabla_\theta \left( loss_\theta^b \right)$;
14:           **end if**
15:       **end for**
16:   **end for**
17:   $i = i + B$
18: **end for**
19: **return** $\pi_{\theta'}(a_t|s_t)$

---

## B  HARDWARE AND CONFIGURATIONS

We use fixed hyperparameters empirically tuned on small problems. We adopt $K = 4$ iterations of TPM and CAM updates. In each TPM iteration, $\text{MLP}_T^k$ has 2 hidden layers with dimension 128,

and $\epsilon^k$ is set to 0 following (Xu et al., 2019). For CAM, both $\text{GAT}_J^k$ and $\text{GAT}_M^k$ have one attention head. For action selection, $\text{MLP}_A$ has 4 hidden layers with dimension 64. All raw features are normalized by dividing a large number, where $p_{ji}$ is divided by 99, and $est_{ji}$ and $lst_{ji}$ are divided by 1000. For each training problem size, we train the policy network with 128000 instances, which are generated on the fly in batches of size 64. We use $n = 10$ and step limit $T = 500$ in our $n$-step REINFORCE (refer to Appendix A for the pseudo-code), with Adam optimizer and constant learning rate $\alpha = 5 \times 10^{-5}$. Another set of 100 instances is generated for validation, which is performed every 10 batches of training. Throughout our experiments, we sample actions from the policy. All initial solutions are computed by a widely used dispatching rule, i.e. the minimum ratio of Flow Due Date to Most Work Remaining (FDD/MWKR) (Sels et al., 2012). During testing, we let our method run for longer improvement steps by setting $T$ to 500, 1000, 2000, and 5000, respectively. Our model is implemented using Pytorch-Geometric (PyG) (Fey & Lenssen, 2019). Other parameters follow the default settings in PyTorch (Paszke et al., 2019). We use a machine with AMD Ryzen 3600 CPU and a single Nvidia GeForce 2070S GPU. We will make our code and data publicly available.

## C COMPUTATIONAL COMPLEXITY ANALYSIS

### C.1 PROOF OF PROPOSITION 4.1

We first prove that the computational complexity for TMP and CAM is linear for the number of jobs $|\mathcal{J}|$ and the number of machines $|\mathcal{M}|$, respectively. Throughout the proofs, we let $|\mathcal{O}| = |\mathcal{J}| \cdot |\mathcal{M}| + 2$ and $|\mathcal{E}| = |\mathcal{C} \cup \mathcal{D}| = |\mathcal{C}| + |\mathcal{D}| = 2|\mathcal{J}| \cdot |\mathcal{M}| + |\mathcal{J}| - |\mathcal{M}|$ denote the total number of nodes and the total number of edges in $G$, respectively.

**Lemma C.1.** *Any layer $k$ of TPM possesses linear computational complexity w.r.t. both $|\mathcal{J}|$ and $|\mathcal{M}|$ when calculating node embedding $\mu_V^k$ and graph level embedding $\mu_G^k$.*

*Proof.* For the $k$-th TPM layer, the matrix form of each MLP layer $\zeta$ can be written as:

$$\mathbf{M}_k = BN^{(\zeta)} \left( \sigma^{(\zeta)} \left( \left( (\mathbf{A} + (1 + \epsilon_k) \cdot \mathbf{I}) \cdot \mathbf{M}_{k-1} \cdot \mathbf{W}_k^{(\zeta)} \right) \right) \right), \text{for } \zeta = 1, \cdots, Z \tag{9}$$

where $\mathbf{M}_{k-1}$, $\mathbf{M}_k \in \mathbb{R}^{|\mathcal{O}| \times p}$ are the node embeddings from layer $k - 1$ and $k$, respectively, $\epsilon_k \in \mathbb{R}$ and $\mathbf{W}_k^{(\zeta)} \in \mathbb{R}^{p \times p}$ are learnable parameters, $\sigma$ is an element-wise activation function (e.g. `relu` or `tanh`), $BN$ is a layer of batch normalization (Ioffe & Szegedy, 2015), and the operator "$\cdot$" denoting the matrix multiplication. Then, the computational time of equation (9) can be bounded by $O(|\mathcal{E}|p^2 + Z|\mathcal{O}|p^2 + Z|\mathcal{O}| \cdot (p^2 + p))$. Specifically, $O(|\mathcal{E}|p^2)$ is the total cost for message passing and aggregation. Note that since we adopt the sparse matrix representation for $\mathbf{A}$, the complexity of message passing and aggregation will be further reduced to $O(|\mathcal{E}|p)$ in practice. The term $Z|\mathcal{O}|p^2$ is the total cost for feature transformation by applying $\mathbf{W}_k^{(\zeta)}$, and $Z|\mathcal{O}| \cdot (p^2 + p)$ is the cost for batch normalization. This analysis shows that the complexity of TPM layer for embedding a disjunctive graph $G$ is linear w.r.t. both $|\mathcal{J}|$ and $|\mathcal{M}|$. Finally, since we read out the graph level embedding by averaging the node embeddings, $\mu_G^k = \frac{1}{|\mathcal{O}|} \sum_{V \in \mathcal{O}} \mu_V^k$, and the final graph-level embedding is just the sum of all that of each layer, i.e. $\mu_G = \sum_k \mu_G^k$, we can conclude that the layer $k$ of TPM possesses linear computational complexity. $\square$

Next, we show that the computational complexity of each CAM layer is also linear w.r.t. both the number of jobs $|\mathcal{J}|$ and the number of machines $|\mathcal{M}|$.

**Lemma C.2.** *Any layer $k$ of CAM possesses linear computational complexity w.r.t both $|\mathcal{J}|$ and $|\mathcal{M}|$ when calculating node embedding $\nu_V^k$.*

*Proof.* Since we employ GAT as the building block for embedding $G_J$ and $G_M$, the computational complexity of the $k$-th layer of CAM is bounded by that of the GAT layer. By referring to the complexity analysis in the original GAT paper (Veličković et al., 2018), it is easy to show that the computational complexity of any layer $k$ of CAM equipped with a single GAT attention head computing $p$ features is bounded by $O(2|\mathcal{O}|p^2 + |\mathcal{E}|p)$, since $G_J$ and $G_M$ both contain all nodes in $G$ but disjoint subsets of edges in $G$ (see Figure 4). Similar to TPM, we employ average read-out function

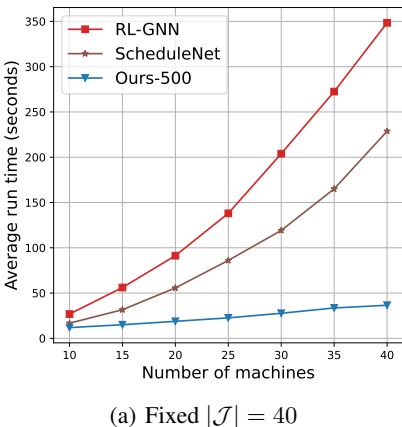 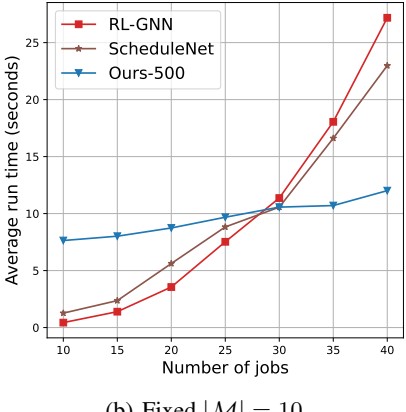

(a) Fixed $|\mathcal{J}| = 40$

(b) Fixed $|\mathcal{M}| = 10$

Figure 5: **The computational complexity of RL-GNN, ScheduleNet, and our method (500 improvement steps).** In the left figure, we fix $|\mathcal{J}| = 40$ and test on various number of machines $|\mathcal{M}|$. While in the right figure, we fix $|\mathcal{M}| = 10$ and test on various number of jobs $|\mathcal{J}|$.

for the final graph level embedding $\nu_G = \frac{1}{|\mathcal{O}|} \sum_{V \in \mathcal{O}} \nu_V^K$, where $\nu_V^K$ is the node embeddings from the last layer $K$. Thus, the overall complexity of CAM is linear w.r.t. both $|\mathcal{J}|$ and $|\mathcal{M}|$. □

Finally, the Theorem 4.1 is proved by the fact that our action selection network is just another MLP whose complexity is linear to the input batch size, which is the total number of nodes in $G$. □

## C.2 COMPUTATIONAL COMPLEXITY COMPARED WITH DRL BASELINES

In this part, we empirically compare the computational complexity with the two best-performing DRL baselines RL-GNN (Park et al., 2021b) and ScheduleNet (Park et al., 2021a). While L2D (Zhang et al., 2020) is very fast, here we do not compare with it due to its relatively poor performance. We conduct two sets of experiments, where we fix one element in the number of jobs $|\mathcal{J}|$ and machines $|\mathcal{M}|$, and change the other to examine the trend of time increase. For each pair of $|\mathcal{J}|$ and $|\mathcal{M}|$, we randomly generate 10 instances and record the average run time.

In Figure 5, we plot the average run time of RL-GNN, ScheduleNet and our method (500 steps) in these two experiments. We can tell that the computational complexity of our method is linear w.r.t. $|\mathcal{J}|$ and $|\mathcal{M}|$, which is in line with our theoretical analysis in Proposition 4.1. In Figure 5(a) where $|\mathcal{J}|$ is fixed, the run time of RL-GNN and ScheduleNet appears to be linearly and quadratically increasing w.r.t $|\mathcal{M}|$, respectively. This is because the number of edges in the graph representations of RL-GNN and ScheduleNet are bounded by $O(|\mathcal{M}|)$ and $O(|\mathcal{M}|^2)$, respectively. While in Figure 5(b) where $|\mathcal{M}|$ is fixed, the run time of RL-GNN and ScheduleNet are quadratically and linearly increasing w.r.t $|\mathcal{J}|$, because the number of edges are now bounded by $O(|\mathcal{J}|^2)$ and $O(|\mathcal{J}|)$, respectively. In Figure 5(b), the reason why our method takes longer time when the problem size is small, e.g. 10×10, is that our method performs 500 improvement steps regardless of the problem size. In contrast, RL-GNN and ScheduleNet are construction methods, for which the construction step closely depends on the problem size. For smaller problems, they usually take less steps to construct a solution, hence are faster. However, when the problem becomes larger, their run time increase rapidly and surpass ours, as shown in the right part of Figure 5(b).

## D MESSAGE-PASSING FOR CALCULATING SCHEDULE

### D.1 PROOF OF THEOREM 4.2

We first show how to compute $est_V$ by the critical path method (CPM) (Jungnickel & Jungnickel, 2005). For any given disjunctive graph $G = (\mathcal{O}, \mathcal{C} \cup \mathcal{D})$ of a solution $s$, there must exist a topological order among nodes of $G$ $\phi : \mathcal{O} \to \{1, 2, \cdots, |\mathcal{O}|\}$, where a node $V$ is said to have the higher order

Table 2: **Run time of our evaluator (MP) versus CPM.** "Speedup" is the ratio of CPM (CPU) to MP (GPU).

| Batch size | 1 | 32 | 64 | 128 | 256 | 512 |
|---|---|---|---|---|---|---|
| MP (CPU) | 0.051s | 0.674s | 1.216s | 2.569s | 5.219s | 10.258s |
| MP (GPU) | 0.058s | 0.094s | 0.264s | 0.325s | 0.393s | 0.453s |
| CPM (CPU) | 0.009s | 0.320s | 0.634s | 1.269s | 2.515s | 5.183s |
| Speedup | | 0.16× | 3.40× | 2.40× | 3.90× | 6.42× | 11.4× |

than node $U$ if $\phi(V) < \phi(U)$. Let $est_S = 0$ for node $O_S$. Then for any node $V \in \mathcal{O}\backslash\{O_S\}$, $est_V$ can be calculated recursively by following the topological order $\phi$, i.e. $est_V = \max_U(p_U + est_U)$ where $U \in \mathcal{N}_V$ is a neighbour of $V$.

*Proof.* Now we show that the message passing is equivalent to this calculation. Specifically, for any node $V$, if $\forall U \in \mathcal{N}_V, c_U = 0$ then $d_V = mp_{max}(\{p_U + (1 - c_U) \cdot d_U | \forall U \in \mathcal{N}(V)\}) = \max_{U \in \mathcal{N}_V}(p_U + d_U)$. Then it is easy to show that the message $c_V = 0$ if and only if $c_W = 0$ for all $W \in \{W | \phi(W) < \phi(V)\}$ and there is a path from $W$ to $V$. This is because of two reasons. First, the message always passes from the higher rank nodes to the lower rank nodes following the topological order in the disjunctive graph $G$. Second, $c_V = 0$ otherwise message $c(W) = 1$ will be recursively passed to node $V$ via the message passing calculation $c_V = mp_{max}(\{c_U | \forall U \in \mathcal{N}(V))$. Hence, $d_U = est_U$ for all $U \in \mathcal{N}_V$ when all $c_U = 0$. Therefore, $d_V = \max_{U \in \mathcal{N}_V}(p_U + d_U) = \max_{U \in \mathcal{N}_V}(p_U + est_U) = est_V$. Finally, $d_T = est_T = C_{max}$ as the message will take up to $H$ steps to reach $O_T$ from $O_S$ and the processing time of $O_T$ is 0. □

### D.2 PROOF OF COROLLARY 4.3

Computing $lst_V$ by CPM follows a similar logic. We define $\bar{\phi} : \mathcal{O} \to \{1, 2, \cdots, |\mathcal{O}|\}$ as the reversed topological order of $V \in \mathcal{O}$, i.e. $\bar{\phi}(V) < \bar{\phi}(U)$ if and only if $\phi(V) > \phi(U)$ (in this case, node $O_T$ will rank the first according to $\bar{\phi}$). Let $lst_T = C_{max}$ for node $O_T$. Then for any node $V \in \mathcal{O}\backslash\{O_T\}$, $lst_V$ can be calculated recursively by following the reversed topological order $\bar{\phi}$, i.e. $lst_V = \min_U(-p_U + lst_U)$ where $V \in \mathcal{N}_U$ is a neighbour of $U$ in $G$.

Now, we can show that $\bar{d}_V = mp_{max}(\{p_U + (1 - \bar{c}_U) \cdot \bar{d}_U\}) = mp_{max}(\{p_U + \bar{d}_U\}) = \max_U(p_U - \bar{d}_U) = \max_U(p_U - lst_U) = -lst_V$ when $\bar{c}_U = 0$ for all $U \in \mathcal{N}_V$ by following the same procedure in the above proof for Proposition 4.2. Finally, $\bar{d}_S = lst_S = 0$ as the message will take up to $H$ steps to reach $O_S$ from $O_T$, since the lengths of the longest path in $G$ and $\bar{G}$ are equal. □

### D.3 EFFICIENCY OF THE MESSAGE-PASSING EVALUATOR

We compare the computational efficiency of our message-passing evaluator with traditional CPM on problems of size $100 \times 20$, which are the largest in our experiments. We implement CPM in Python, following the procedure in Section 3.6 of (Jungnickel & Jungnickel, 2005). We randomly generate batches of instances with size 1, 32, 64, 128, 256 and 512, and randomly create a directed disjunctive graph (i.e. a solution) for each instance. We gather the total run time of the two methods in Table 2, where we report results of both the CPU and GPU versions of our evaluator. We can observe that although our message-passing evaluator does not present any advantage on a single graph, the performance drastically improves along with the increase of batch size. For a batch of 512 graphs, our evaluator runs on GPU is more than 11 times faster than CPM. We can conclude that our message-passing evaluator with GPU is superior in processing large graph batches, confirming its effectiveness in utilizing the parallel computational power of GPU. This offers significant advantage for deep (reinforcement) learning based scheduling methods, in terms of both the training and testing phase (for example, solving a batch of instances, or solving multiple copies of a single instance in parallel and retrieving the best solution).

### D.4 AN EXAMPLE OF CALCULATING *est* BY USING THE PROPOSED MESSAGE-PASSING EVALUATOR.

In Figure 6 we present an example of calculating the earliest starting time for each operation in a disjunctive graph using our proposed message-passing operator.

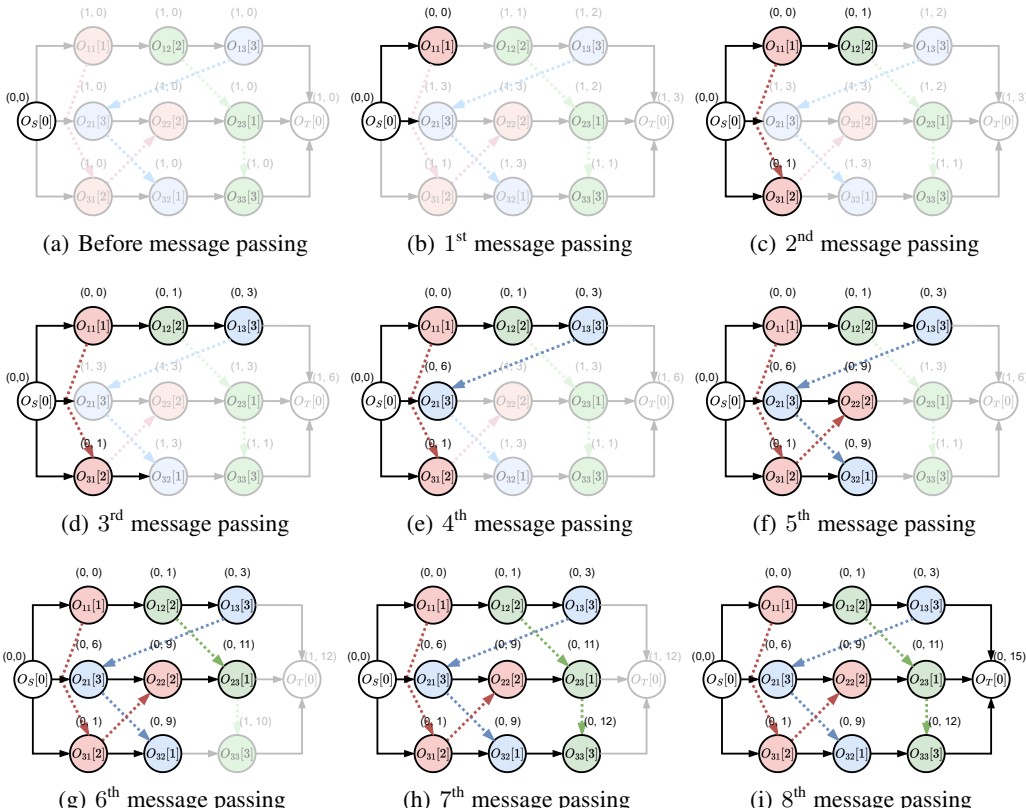

Figure 6: **An example of the forward message passing for calculating** $est$**.** The pair of numbers $(d_V, c_V)$ in the open bracket and the number in the square bracket $[p_V]$ denote the messages and the processing time for each vertex $V$, respectively. In each graph, we highlight the nodes $V$ with $c_V = 0$. After eight times of message passing, $c_V$ for any node $V$ equals 0, which can be utilized as a signal for termination. Then, the message $d_V$ equals the earliest starting time $est_V$ for each $V$. It is clear that the frequency of message passing (8) is less than $H$, the length of the longest path containing the most nodes, which is 9 $(O_S, O_{11}, O_{12}, O_{13}, O_{21}, O_{22}, O_{23}, O_{33}, O_T)$.

## E HAND-CRAFTED IMPROVEMENT RULES

Here we present details of the three hand-crafted rule baselines, i.e. greedy (GD), best-improvement (BI), and first-improvement (FI). The greedy rule is widely used in the tabu search framework for JSSP (Zhang et al., 2007). The best-improvement and the first-improvement rules are widely used in solving combinatorial optimization algorithms (Hansen & Mladenović, 2006). Specifically, the greedy rule selects the solution with the smallest makespan in the neighbourhood, while the best-improvement and the first-improvement rules select the best makespan-reducing and the first makespan-reducing solution in the neighbourhood, respectively. If there is no solution in the neighbourhood with a makespan lower than the current one, the best-improvement and first-improvement rules cannot pick any solutions. However, the greedy rule will always have solutions to pick.

It is unfair to directly compare with BI and FI since they may stuck in the local minimum when they cannot pick any solutions. Thus we augment them with *restart*, a simple but effective strategy widely used in combinatorial optimization, to pick a new starting point when reaching local minimum

(Lourenço et al., 2019). While random restart is a simple and direct choice, it performs poorly probably because the solution space is too large (Lourenço et al., 2019). Hence, we use a more advanced restart by adopting a memory mechanism similar to the one in tabu search (Zhang et al., 2007). Specifically, we maintain a long-term memory $\Omega$ which keeps tracking the $\omega$ latest solutions in the search process, from which a random solution is sampled for restart when reaching local minimum. The memory capacity $\omega$ controls the exploitation and exploration of the algorithm. If $\omega$ is too large, it could harm the performance since bad solutions may be stored and selected. On the contrary, too small $\omega$ could be too greedy to explore solutions that are promising. Here we set $\omega = 100$ for a balanced trade-off. We do not equip GD with restart since it always has solutions to select unless reaching the absorbing state.

## F COMPARISON WITH THE ONLINE DRL-BASED METHOD

The makespan of our method (trained on size 20×15) with 5000 improvement steps and the online DRL-based method (Tassel et al., 2021) in solving the Taillard 30×20 instances are shown in Table 3. Note that this method performs training for each instance individually, while our method learns only one policy offline without specific tuning for each instance.

Table 3: **Detailed makespan values compared with the online DRL method**

| Method | Tai41 | Tai42 | Tai43 | Tai44 | Tai45 | Tai46 | Tai47 | Tai48 | Tai49 | Tai50 |
|---|---|---|---|---|---|---|---|---|---|---|
| (Tassel et al., 2021) | **2208** | **2168** | **2086** | 2261 | 2227 | 2349 | **2101** | 2267 | **2154** | 2216 |
| Ours-5000 | 2248 | 2186 | 2144 | **2202** | **2180** | **2291** | 2123 | **2167** | 2167 | **2188** |

## G TRAINING CURVES

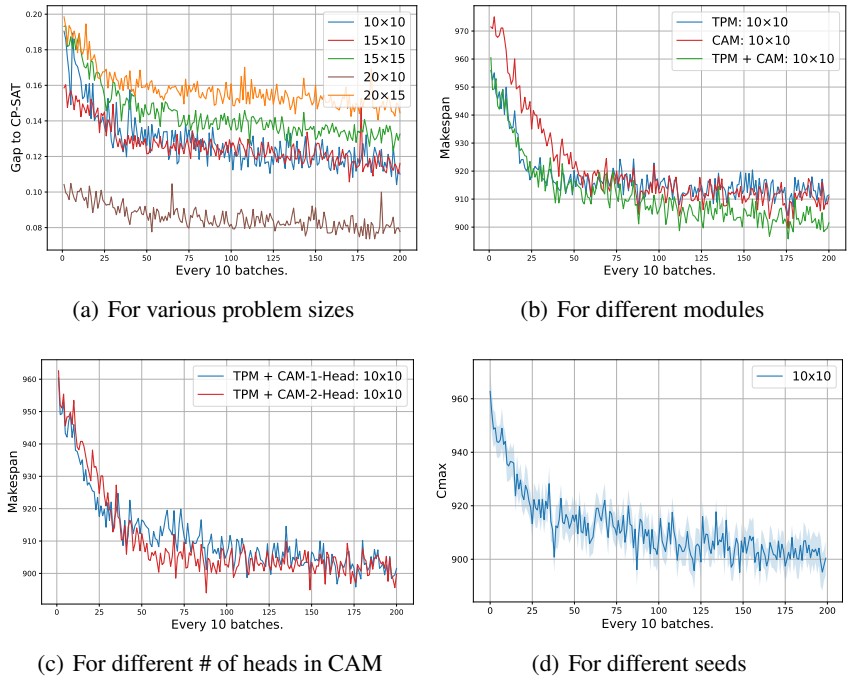

(a) For various problem sizes

(b) For different modules

(c) For different # of heads in CAM

(d) For different seeds

Figure 7: **Training curves.**

# H    COMPARISON WITH TABU SEARCH

We compare our method with a Tabu search-based metaheuristic algorithm with the dynamic tabu size proposed in (Zhang et al., 2007). Since the neighbourhood structure in (Zhang et al., 2007) is different from $N_5$, to make the comparison fair, we replace it with $N_5$ (TSN5). We also equip the tabu search method with our message-passing evaluator to boost speed. We test on all seven public datasets, where we conduct two experiments. In the first experiment, we fix the search steps to 5000. In the second one, we adopt the same amount of computational time of 90 seconds (already sufficient for our method to generate competitive results) for both methods. The results for these two experiments are presented in Table 4 and Table 5, respectively.

Table 4 indicates that our method closely trails Tabu search with a 1.9% relative gap. This is due to the simplicity of our approach as a local search method without complex specialized mechanisms (Figure 2), making direct comparison less equitable. However, our method is significantly faster than Tabu search since it avoids evaluating the entire neighbourhood for move selection. Conversely, Table 5 reveals that our method outperforms Tabu search under the same time constraint. This is attributed to the desirable ability of our method to explore the solution space more efficiently within the allotted time, as it does not require a full neighbourhood evaluation for move selection.

Table 4: **Performance compared with Tabu search for 5000 improvement steps.** For each problem size, we compute the average relative gap of the makespan of our method to the tabu search algorithm, and we report the time (in seconds) for each method for solving a single instance.

| Method | Taillard | | | | | | | | ABZ | | FT | | |
|---|---|---|---|---|---|---|---|---|---|---|---|---|---|
| | $15 \times 15$ | $20 \times 15$ | $20 \times 20$ | $30 \times 15$ | $30 \times 20$ | $50 \times 15$ | $50 \times 20$ | $100 \times 20$ | $10 \times 10$ | $20 \times 15$ | $6 \times 6$ | $10 \times 10$ | $20 \times 5$ |
| | Gap Time | Gap Time | Gap Time | Gap Time | Gap Time | Gap Time | Gap Time | Gap Time | Gap Time | Gap Time | Gap Time | Gap Time | Gap Time |
| TSN5 | 0.0% 271.5s | 0.0% 311.4s | 0.0% 369.4s | 0.0% 378.6s | 0.0% 422.5s | 0.0% 547.9s | 0.0% 573.0s | 0.0% 1045.8s | 0.0% 177.8s | 0.0% 283.8s | 0.0% 124.0s | 0.0% 213.1s | 0.0% 235.4s |
| Ours | 2.8% 92.2s | 3.8% 102.1s | 3.2% 114.3s | 2.8% 120.7s | 4.1% 144.4s | 3.5% 168.7s | 2.7% 228.1s | 1.5% 504.6s | -0.1% 75.2s | 3.1% 99.6s | 0.0% 67.7s | 1.7% 74.8s | 0.4% 73.3s |

| Method | LA | | | | | | | | SWV | | | ORB | YN |
|---|---|---|---|---|---|---|---|---|---|---|---|---|---|
| | $10 \times 5$ | $15 \times 5$ | $20 \times 5$ | $10 \times 10$ | $15 \times 10$ | $20 \times 10$ | $30 \times 10$ | $15 \times 15$ | $20 \times 10$ | $20 \times 15$ | $50 \times 10$ | $10 \times 10$ | $20 \times 20$ |
| | Gap Time | Gap Time | Gap Time | Gap Time | Gap Time | Gap Time | Gap Time | Gap Time | Gap Time | Gap Time | Gap Time | Gap Time | Gap Time |
| TSN5 | 0.0% 56.5s | 0.0% 81.9s | 0.0% 116.8s | 0.0% 192.4s | 0.0% 225.1s | 0.0% 233.4s | 0.0% 149.5s | 0.0% 263.1s | 0.0% 343.1s | 0.0% 383.4s | 0.0% 719.9s | 0.0% 225.0s | 0.0% 318.2s |
| Ours | 1.4% 70.0s | 0.0% 71.0s | 0.0% 73.7s | -0.3% 75.1s | 0.5% 80.9s | 0.8% 85.4s | 0.0% 99.3s | 3.1% 88.8s | 2.7% 86.9s | 3.9% 99.8s | 4.1% 126.3s | 1.1% 75.9s | 3.6% 113.2s |

Table 5: **Performance compared with Tabu search for two minutes.** We report the average gap to the upper bound solution.

| Method | Taillard | | | | | | | | ABZ | | FT | | |
|---|---|---|---|---|---|---|---|---|---|---|---|---|---|
| | $15 \times 15$ | $20 \times 15$ | $20 \times 20$ | $30 \times 15$ | $30 \times 20$ | $50 \times 15$ | $50 \times 20$ | $100 \times 20$ | $10 \times 10$ | $20 \times 15$ | $6 \times 6$ | $10 \times 10$ | $20 \times 5$ |
| Ours | 5.9% | 7.9% | 8.8% | 9.0% | 12.9% | 4.9% | 7.3% | 6.2% | 1.2% | 8.1% | 0.0% | 5.3% | 0.7% |
| TSN5 | 6.1% | 8.0% | 8.8% | 9.3% | 12.4% | 5.0% | 7.4% | 6.5% | 1.6% | 8.3% | 0.0% | 5.6% | 1.2% |

| Method | LA | | | | | | | | SWV | | | ORB | YN |
|---|---|---|---|---|---|---|---|---|---|---|---|---|---|
| | $10 \times 5$ | $15 \times 5$ | $20 \times 5$ | $10 \times 10$ | $15 \times 10$ | $20 \times 10$ | $30 \times 10$ | $15 \times 15$ | $20 \times 10$ | $20 \times 15$ | $50 \times 10$ | $10 \times 10$ | $20 \times 20$ |
| Ours | 0.4% | 0.0% | 0.0% | 0.4% | 2.7% | 2.1% | 0.0% | 5.3% | 13.4% | 14.7% | 17.1% | 2.7% | 8.5% |
| TSN5 | 0.0% | 0.0% | 0.0% | 0.9% | 3.0% | 2.3% | 0.0% | 5.4% | 13.4% | 14.8% | 17.3% | 3.0% | 8.8% |

# I    ENSEMBLE PERFORMANCE

The ensemble results for testing and generalization on the seven benchmarks are presented in Table 6. For each instance, we simply run all the five trained policies and retrieve the best solution. We can observe that for most of the cases, with the same improvement steps, the ensemble strategy can further improve the performance.

# J    STATE TRANSITION

The state transition is highly related to the $N_5$ neighbourhood structure. As we described in Preliminaries (Section 3), once our DRL agent selects the operation pair, say $(O_{13}, O_{21})$ in the sub-figure on the right of Figure 1, the $N_5$ operator will swap the processing order of $O_{13}$ and $O_{21}$. Originally, the processing order between $O_{13}$ and $O_{21}$ is $O_{13} \rightarrow O_{21}$. After swapping, the processing order will become $O_{13} \leftarrow O_{21}$. However, solely changing the orientation of arc $O_{13} \rightarrow O_{21}$ is not enough. In this case, we must break the arc $O_{21} \rightarrow O_{32}$ and introduce a new disjunctive arc $O_{13} \rightarrow O_{32}$ since

Table 6: **Ensemble results on classic benchmarks.** Values in the table are the average gap to the best solutions in the literature. **Bold** means the ensemble strategy performs better with the same improvement steps.

| Method | Taillard | | | | | | | | ABZ | | FT | | |
|---|---|---|---|---|---|---|---|---|---|---|---|---|---|
| | 15×15 | 20×15 | 20×20 | 30×15 | 30×20 | 50×15 | 50×20 | 100×20 | 10×10 | 20×15 | 6×6 | 10×10 | 20×5 |
| Closest-500 | 9.3% | 11.6% | 12.4% | 14.7% | 17.5% | 11.0% | 13.0% | 7.9% | 2.8% | 11.9% | 0.0% | 9.9% | 6.1% |
| Closest-1000 | 8.6% | 10.4% | 11.4% | 12.9% | 15.7% | 9.0% | 11.4% | 6.6% | 2.8% | 11.2% | 0.0% | 8.0% | 3.9% |
| Closest-2000 | 7.1% | 9.4% | 10.2% | 11.0% | 14.0% | 6.9% | 9.3% | 5.1% | 2.8% | 9.5% | 0.0% | 5.7% | 1.5% |
| Closest-5000 | 6.2% | 8.3% | 9.0% | 9.0% | 12.6% | 4.6% | 6.5% | 3.0% | 1.4% | 8.6% | 0.0% | 5.6% | 1.1% |
| Ensemble-500 | **8.8%** | 11.6% | 12.0% | **14.4%** | 17.5% | **10.4%** | **12.9%** | 7.6% | **2.4%** | **11.5%** | 0.0% | **8.5%** | 6.1% |
| Ensemble-1000 | **6.3%** | 10.4% | **10.9%** | **12.2%** | 15.7% | **8.5%** | 11.0% | **6.2%** | **1.7%** | **10.6%** | 0.0% | 8.0% | 3.9% |
| Ensemble-2000 | **5.9%** | **9.1%** | **9.6%** | **10.6%** | 14.0% | **6.5%** | **9.2%** | **4.3%** | **1.7%** | **9.4%** | 0.0% | 5.7% | 1.5% |
| Ensemble-5000 | **5.5%** | **8.0%** | **8.6%** | **8.5%** | 12.4% | **4.1%** | 6.5% | **2.3%** | **0.8%** | 8.5% | 0.0% | **4.7%** | 1.1% |

| Method | LA | | | | | | | | SWV | | | ORB | YN |
|---|---|---|---|---|---|---|---|---|---|---|---|---|---|
| | 10×5 | 15×5 | 20×5 | 10×10 | 15×10 | 20×10 | 30×10 | 15×15 | 20×10 | 20×15 | 50×10 | 10×10 | 20×20 |
| Closest-500 | 2.1% | 0.0% | 0.0% | 4.4% | 6.4% | 7.0% | 0.2% | 7.3% | 29.6% | 25.5% | 21.4% | 8.2% | 12.4% |
| Closest-1000 | 1.8% | 0.0% | 0.0% | 2.3% | 5.1% | 5.7% | 0.0% | 6.6% | 24.5% | 23.5% | 20.1% | 6.6% | 10.5% |
| Closest-2000 | 1.8% | 0.0% | 0.0% | 1.8% | 4.0% | 3.4% | 0.0% | 6.3% | 21.8% | 21.7% | 19.0% | 5.7% | 9.6% |
| Closest-5000 | 1.8% | 0.0% | 0.0% | 0.9% | 3.4% | 2.6% | 0.0% | 5.9% | 17.8% | 17.0% | 17.1% | 3.8% | 8.7% |
| Ensemble-500 | 2.1% | 0.0% | 0.0% | **3.0%** | **4.7%** | **6.9%** | **0.1%** | 7.3% | **27.2%** | 25.5% | 21.4% | **8.0%** | 12.4% |
| Ensemble-1000 | **1.6%** | 0.0% | 0.0% | 2.3% | **3.9%** | 5.7% | 0.0% | 6.6% | 24.5% | 23.5% | **19.9%** | **6.5%** | 10.5% |
| Ensemble-2000 | **1.6%** | 0.0% | 0.0% | 1.8% | **3.9%** | 3.4% | 0.0% | 6.2% | **21.7%** | 21.7% | **18.9%** | 5.4% | 9.4% |
| Ensemble-5000 | **1.6%** | 0.0% | 0.0% | 0.9% | 3.4% | 2.6% | 0.0% | **5.0%** | 17.8% | 17.0% | 17.1% | 3.8% | **7.4%** |

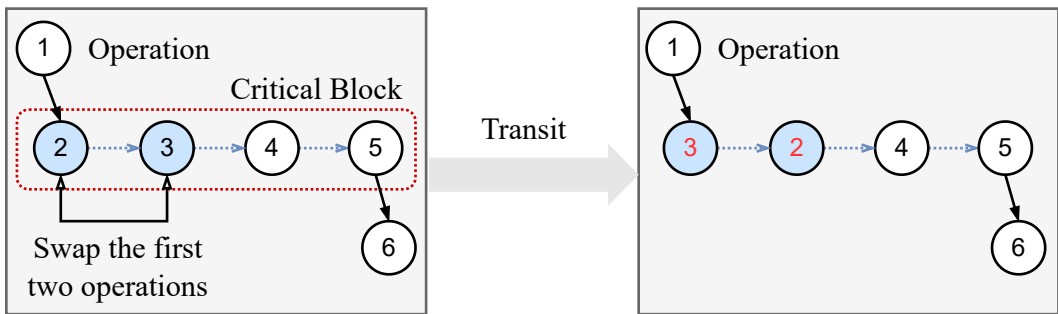

Figure 8: **An example of state transition using $N_5$ neighbourhood structure.**

the operation $O_{32}$ is now the last operation to be processed on the blue machine. In some cases, the state transition involves several steps. For example, in Figure 8 below, given a processing order $O_2 \to O_3 \to O_4 \to O_5$ on some machine, if we swap $O_2$ and $O_3$ ($O_2$ and $O_3$ are the first two operations of some critical block), we need to break disjunctive arcs $O_1 \to O_2$ and $O_3 \to O_4$, flip the orientation of arc $O_2 \to O_3$ as $O_2 \leftarrow O_3$, finally, add new arcs $O_3 \to O_1$ and $O_2 \to O_4$. So, the new processing order after the transition will be $O_1 \to O_3 \to O_2 \to O_4$ (right figure).

In summary, after the transition, the new disjunctive graph (state) will have a different processing order for the machine assigned to the selected operation pair, but other machines stay unaffected.

Then, the starting time (schedule) of every operation is re-calculated according to the proposed message-passing evaluator (Theorem 4.2).

## K   THE TOPOLOGICAL RELATIONSHIPS COULD BE MORE NATURALLY MODELLED AMONG OPERATIONS IN THE DISJUNCTIVE GRAPH.

A disjunctive graph effectively represents solutions by using directed arrows to indicate the order of processing operations, with each arrow pointing from a preceding operation to its subsequent one. The primary purpose of the disjunctive graph is to encapsulate the topological relationships among the nodes, which are defined by these directional connections. Thus, the disjunctive graph excels

in illustrating the inter-dependencies between operations, in contrast to other visualization methods, such as Gantt charts, which are better suited for presenting explicit scheduling timelines.

In formulating partial solutions within a construction step, the agent must possess accurate work-in-progress information, which encompasses specifics such as the extant load on each machine and the current status of jobs in process. The challenge encountered in this scenario relates to the difficulty in identifying suitable entities within the disjunctive graph that can store and convey such information effectively. While it is conceivable to encode the load of a machine using numerical values, no discrete node within the disjunctive graph explicitly embodies this information. Instead, a machine's representation is implicit and discerned through the collective interpretation of the operations it processes, effectively forming what is termed a 'machine clique.' This indirect representation necessitates a more nuanced approach to ensuring that the agent obtains the necessary information to construct viable partial solutions.

## L    RESULT FOR EXTREMELY LARGE INSTANCES

To comprehensively evaluate the generalization performance of our method, we consider another three problem scales (up to 1000 jobs), namely $200 \times 40$ (8,000 operations), $500 \times 60$ (30,000 operations), and $1000 \times 40$ (40,000 operations). We randomly generate 100 instances for each size and report the average gap to the CP-SAT. Hence, the negative gap indicates the magnitude in percentage that our method outperforms CP-SAT. We use our model trained with size $20 \times 15$ for comparison. The result is summarized in Table 7.

Table 7: **Generalization performance on extremely large datasets.**

|          | 200x40        | 500x60        | 1000x40        |
|----------|---------------|---------------|----------------|
| CP-SAT   | 0.0% (1h)     | 0.0% (1h)     | 0.0% (1h)      |
| Ours-500 | -24.31% (88.7s) | -20.56% (3.4m) | -15.99% (4.1m) |

From the results in the table, our method shows its advantage against CP-SAT by outperforming it for the extremely large-scale problem instances, with only 500 improvement steps. The number in the bracket is the average time of solving a single instance, where "s", "m", and "h" denote seconds, minutes, and hours, respectively. The computational time can be further reduced by batch processing.

## M    ABLATION STUDIES

In Figure 7(a) and Figure 7(d) (Appendix G), we display the training curves of our method on all five training problem sizes and for three random seeds on size $10 \times 10$, respectively. We can confirm that our method is fairly reliable and stable for various problem sizes and different random seeds. We further conduct an ablation study on the architecture of the policy network to verify the effectiveness of combining TPM and CAM in learning embeddings of disjunctive graphs. In Figure 7(b), we display the training curve of the original policy network, as well as the respective ones with only TPM or CAM, on problem size $10 \times 10$. In terms of convergence, both TPM and CAM are inferior to the combination, showing that they are both important in extracting useful state features. Additionally, we also analyze the effect of different numbers of attention heads in CAM, where we train policies with one and two heads while keeping the rest parts the same. The training curves in Figure 7(c) show that the policy with two heads converges faster. However, the converged objective values have no significant difference.

