# OpenReview forum: "Deep Reinforcement Learning Guided Improvement Heuristic for Job Shop Scheduling"
_ICLR.cc/2024/Conference — ICLR 2024 poster_

### Official Review · Reviewer_W4mB · 2023-10-31

**Soundness:** 3 good
**Presentation:** 3 good
**Contribution:** 3 good
**Rating:** 8
**Confidence:** 3

**Summary:**

The paper proposes a novel, learning-based improvement heuristic for the job shop scheduling problem. It uses a novel graph neural network  to select operations that modify a solution from a set of candidate operations defined by the N5 neighborhood structure. The novel network architecture consists of a topology embedding module and a context-aware embedding module that are designed to encode the problem instances and their solutions. The authors prove that their method scales linearly with the problem size. Furthermore, they evaluate their method on a variety of instances from the literature and compare it to state-of-the-art ML methods, simple handcrafted heuristics, and a tabu search using the same neighborhood structure. Overall, the proposed method shows good performance outperforming existing ML methods in most settings.

**Strengths:**

- The proposed method introduces several novel concepts. In contrast, to many other approaches it is an improvement heuristic (instead of a construction heuristic). The novel network architecture is specifically designed to encode existing solutions and consists of two components that both are independently evaluated in an ablation study and seem to be well suited for the task. Furthermore, the REINFORCE algorithm has been slightly adapted for the task at hand.
- The method scales linearly with the problem size. In contrast, to many other machine learning based approaches the proposed method can thus scale to larger problem instances.
- The proposed method outperforms other state-of-the-art learning-based approaches for the job shop scheduling problem.
- Overall, the paper is well written.
- The authors conduct ablation studies that evaluate individual components of the method.

**Weaknesses:**

- The method uses the handcrafted N5 neighborhood to generate candidate operations (i.e., the set of possible actions for the model). This means that the method relies on a handcrafted component specific to the job shop scheduling problem. It is thus unclear how easily the method can be modified to solve other scheduling problems. Despite being based on a handcrafted heuristics it does not outperform the handcrafted tabu search using the same neighborhood structure in all settings.
- The authors only evaluate their method on the job shop scheduling problem and not on any other scheduling problems.

**Questions:**

- Have you experimented with different ways to calculate the reward? For your current method, the network might receive different rewards when picking the same actions at the same state depending on what solutions were found in previous states. While I understand the motivation of your current choice, I wonder if this might impact learning.
- Have you tried to use a baseline during learning for REINFORCE?

---

> ### Author Response · Authors · 2023-11-19
> **Response to Reviewer W4mB**
>
> **C1: It is thus unclear how easily the method can be modified to solve other scheduling problems.**
>
> **A**: We want to take the opportunity to illustrate that our method can be applied to other variants of shop scheduling problems, such as flexible job-shop scheduling problems (FJSSP) and flow shop scheduling problems (FSSP). Such potential is due to the main building blocks of our method, i.e. disjunctive graph-based state representation and critical block-based neighbourhood structure, which are central concepts in various shop scheduling problems. Below we take FJSSP as an example, which is an extension of JSSP where each operation can be processed by one of a set of eligible machines. Selecting machines for the operations introduces extra complexity. Thus, FJSSP is considered more challenging than JSSP. However, we will show that FJSSP can be solved by applying the proposed method with minor modifications. Specifically, the neighbourhood search-based heuristic for FJSSP also depends on the disjunctive graph. For FJSSP, the action will be to reinsert an operation from its current machine into another eligible one (or a different position in the same machine) such that the resulting disjunctive graph contains no cycles, i.e., a DAG. In other words, the action removes the selected operation (and all its arcs) and then inserts it into another position (connecting all necessary arcs) in the disjunctive graph. The optional operations and their feasible insertion positions are defined by some neighbourhood structure, e.g., $N opt1$ and $N opt2$ proposed in the paper [AR3]. In this case, our method could learn to pick the operations and the corresponding reinsertion positions within these neighbourhood structures. To this end, the action set and state transition of the MDP modelling should be modified accordingly, while other components, e.g., the state and reward, could stay the same. Furthermore, our proposed GNN model should remain effective and preserve the linear computational complexity since the state is the disjunctive graph, the same as the one in the manuscript (to see this, compare section 2 of paper [AR3] with our disjunctive graph representation). We will thoroughly investigate expanding our method to FJSSP in the future. For FSSP, our method can be directly applied without changing the neighbourhood structure. Therefore, we quickly built a prototype and evaluated our method on FSSP. In the table below, we use three different scales of FSSP instances from the Taillard dataset [AR4] as the evaluation sets, where we consider four baselines, i.e., CP-SAT (3600s), BI, FI, and GD. For each baseline method, we compute the relative gap between the baseline and the best upper bound reported in the literature. Our method is trained independently for each size with randomly generated instances from the uniform distribution that is usually different from those generating instances in the Taillard dataset [AR4], which indicates we also test the **zero-shot** generalization performance of our method. From the experiment results, we can observe that the local operator learned by our method preserves the superiority to the hand-crafted ones, and the relative gap of the performance to CP-SAT is small. The number in the bracket is the average time of solving a single instance, where “s”, “m”, and “h” denote seconds, minutes, and hours, respectively.
>
> |           |  Tai-FSSP 20x5 | Tai-FSSP 20x10 |  Tai-FSSP 20x20 |
> |:---------:|:--------------:|:--------------:|:---------------:|
> |   CP-SAT  |  0.0% (46.3s)  |   0.01% (1h)   |   0.02% (1h)    |
> |  GD-5000  |  3.03% (5.3m)  | 14.32% (8.0m)  | 17.89% (18.3m)  |
> |  FI-5000  |  2.78% (6.1m)  | 12.25% (9.3m)  | 15.31% (22.1m)  |
> |  BI-5000  |  2.69% (5.7m)  | 12.79% (9.9m)  | 16.26% (24.5m)  |
> | Ours-5000 | 1.72% (72.0s)  | 9.24% (83.2s)  |  13.14% (1.8m)  |
>
> [AR3] Mastrolilli, M., \& Gambardella, L. M. (2000). Effective neighborhood functions for the flexible job shop problem. *Journal of scheduling*, 3(1), 3-20.
>
> [AR4] Eric Taillard. Benchmarks for basic scheduling problems. *European Journal of Operational Research*, 64(2):278–285, 1993.

---

> ### Author Response · Authors · 2023-11-19
> **Response to Reviewer W4mB cnt.**
>
> **C2: The authors only evaluate their method on the job shop scheduling problem and not on any other scheduling problems.**
>
> **A**: Please refer to the response of C1.
>
> **C3: Have you experimented with different ways to calculate the reward?**
>
> **A**: We agree that the success of reinforcement learning algorithms is fundamentally interlinked with the judicious design of reward structures, as various reward configurations can precipitate distinct learning outcomes. In alignment with our aim to incrementally elevate solution quality, we have crafted a reward strategy that inherently encourages continual progress by providing rewards based on the relative improvement over a benchmark incumbent solution. Contrarily, our experiments with a reward model premised on consecutive state reward differences ($r_{t+1} - r_t$) yielded unpredictable learning behaviours, which could be ascribed to the absence of a stable comparative reference. Such findings accentuate the critical role of reward scheme selection in shaping the reinforcement learning process.
>
> **C4: Have you tried to use a baseline during learning for REINFORCE?**
>
> **A**: As witnessed in many previous works, incorporating a baseline into the REINFORCE algorithm to enhance training stability is promising and warrants in-depth investigation. Notwithstanding, selecting appropriate baseline functions for our proposed n-step REINFORCE framework proves nontrivial, mainly due to the policy updates that transpire after every n steps of Monte Carlo sampling. This difficulty is further accentuated within the realm of continuing or non-episodic Markov Decision Processes (MDPs) — as the MDP our paper proposes — where conventional methods of computing the baseline at episode termination cannot be applied. Currently, our method shows good performance without using a baseline. Nevertheless, we will follow your advice and investigate how to design a proper baseline to enhance our method further.

---

### Official Review · Reviewer_Yw7H · 2023-10-31

**Soundness:** 3 good
**Presentation:** 3 good
**Contribution:** 3 good
**Rating:** 8
**Confidence:** 3

**Summary:**

This work addresses the job shop scheduling problem (JSSP), a classic combinatorial optimization task. It proposes a solution based on deep reinforcement learning to learn improvement heuristics, which iteratively refines an existing solution. The authors leverage a disjoint graph representation of the state (which is more complete than that employed in prior works), and propose a GNN architecture for its representation. The proposed MDP formulation chooses a pair of operations to be swapped, and the RL model is trained via the policy gradient. An experimental evaluation is presented, which compares the method with a large set of prior approaches, spanning deep RL, metaheuristic, and exact methods. The evaluation shows favorable results in comparison with these baselines.

**Strengths:**

**S1**. The paper addresses a highly practical problem that has received comparatively less attention (e.g., compared with routing problems) in the combinatorial optimization literature. The proposed method is fairly original, and is similar in spirit with other learned improvement heuristics.

**S2**. The evaluation performed by the authors is very comprehensive (in terms of datasets, instance sizes, baselines, and scenarios). Most importantly, it shows that the method attains excellent performance while requiring a reasonable time budget. It is one of the few cases where opting for the ML model over a classic solver or metaheuristic may actually be preferable in practice.

**Weaknesses:**

**W1**. The only issues that I have identified relate to the clarity of the presentation of the problem formulation and solution method. More specifically:

- The problem formulation uses two different semantics for the $ji$ notation ($O_{ji}$ and $m_{ji}$). A different subscript is needed to indicate the machine, and it should be reflected in the figures as well.
- I was not able to fully understand the transitions induced by the swapping of the two operations, given that it appears to trigger other reconfigurations of the disjunctive graph beyond the two vertices indicated by the action (e.g., the changes to the edges connecting the red machine in Figure 2b). How are these other reconfigurations determined? This further detail should be added in Section 4.1.

**Questions:**

**C1**. There are some typos / informal language / awkward word choices in the manuscript, and it would benefit from further revisions until publication. Some examples:

- "receives relatively less attention" -> received / has received (p1)
- "In specific" -> specifically (p1)
- "A delicate Markov decision process formulation" (p2)
- "graph-based ones aforementioned" -> aforementioned graph-based ones
- "Precedent constraints" -> precedence constraints (p3)
- "To fast calculate" -> to quickly calculate (p3)
- "till" -> until (p4)

**C2**. The results in Appendix H / Table 4 appear very positive and show good generalization performance and scalability. I think it is worth making room for them in the main text.

**C3**. "Parameter $\theta$" -> Parameter *set* $\theta$ in 4.2?

---

> ### Author Response · Authors · 2023-11-19
> **Response to Reviewer Yw7H**
>
> **C1: A different subscript is needed to indicate the machine, and it should be reflected in the figures as well.**
>
> **A**: We sincerely appreciate your valuable feedback and have taken steps to enhance the manuscript accordingly. Please refer to the updated manuscript for details.
>
> **C2: I was not able to fully understand the transitions induced by the swapping of the two operations**
>
> **A**: This question is related to the C2 raised by reviewer SV6I. For convenience, we explain in detail with examples how the swapping of the two operations works.
>
> When our deep reinforcement learning (DRL) agent selects a pair of operations, such as $(O_{13}, O_{21})$ depicted on the right side of Figure 1, it engages the $N_5$ operator to interchange the sequence in which $O_{13}$ and $O_{21}$ are processed. Initially, the sequence $O_{13} \rightarrow O_{21}$ is established. Post-swap, the sequence is revised to $O_{13} \leftarrow O_{21}$. This operation, however, is insufficient in isolation. Specifically, in our example, it is necessary to break the arc from $O_{21}$ to $O_{32}$ and establish a new disjunctive arc from $O_{13}$ to $O_{32}$, given that $O_{32}$ succeeds as the terminal operation on the specified blue machine.
>
> At times, the state transition may invoke a multi-step process. For instance, given a sequential order of operations on a particular machine such as $O_{1} \rightarrow O_{2} \rightarrow O_3 \rightarrow O_4$, and a swap between $O_{2}$ and $O_{3}$ within a critical block, multiple arc modifications are required. This involves removing the arcs from $O_{1}$ to $O_{2}$ and from $O_{3}$ to $O_{4}$, inverting the orientation of the arc from $O_{2}$ to $O_{3}$ to render it $O_{2} \leftarrow O_{3}$, and instituting new arcs from $O_1$ to $O_3$ and from $O_2$ to $O_4$. Consequently, the new processing sequence after the transition is finalized as $O_{1} \rightarrow O_{3} \rightarrow O_2 \rightarrow O_4$.
>
> To summarize, the transitioned state results in an updated disjunctive graph displaying an altered processing sequence on the machine associated with the chosen operational pair, while the sequences on all other machines remain unchanged.
>
> **C3: There are some typos / informal language / awkward word choices in the manuscript, and it would benefit from further revisions until publication.**
>
> **A**: We sincerely appreciate your valuable feedback. Following your suggestions, we will meticulously revisit and revise our manuscript to ensure its utmost quality until publication.
>
> **C4: The results in Appendix H / Table 4 appear very positive and show good generalization performance and scalability. I think it is worth making room for them in the main text.**
>
> **A**: We have updated it in the manuscript.
>
> **C5: "Parameter $\theta$" $->$ Parameter set $\theta$ in 4.2?**
>
> **A**: We thank the reviewer for pointing out the typo and have updated the changes in the manuscript.

---

### Official Review · Reviewer_jCmi · 2023-11-01

**Soundness:** 3 good
**Presentation:** 3 good
**Contribution:** 3 good
**Rating:** 8
**Confidence:** 4

**Summary:**

The paper presents a local search method based on deep reinforcement learning (RL) for the job shop scheduling problem. The authors propose to encode solutions as disjunctive graphs and design a novel graph neural network architecture to process them. In addition, they propose an efficient technique to compute the schedule of a solution on GPU.

**Strengths:**

A novel graph neural network architecture is proposed to process the different type of information encoded in a disjunctive graph: graph topology, operation order in the solution, and operation order on a machine.

An efficient method to compute the schedules of batch of solutions is proposed, which can accelerate training deep RL training.

The authors performed a broad series of experiments on various instances comparing their proposed method with other deep RL algorithms, some basic heuristic methods, but also a meta-heuristic method (tabu search).

**Weaknesses:**

The presentation is generally clear. However, some parts could be improved.

I believe that some notions could be explained in a better way. For instance:
- page 1: "given that the topological relationships could be more naturally modeled among operations" is not clear to me.
- page 3: the notion of N5 neighborhood could be more explained more formally (instead of the current text and the example).
- Some sentences/expressions are unclear (see Questions).

Section 4.2.1: The first two sentences seem to contradict themselves.
What is meant by "machine predecessor" becomes only clear later on.

Minor:
page 7: a closing curly brace is missing for c_V.

**Questions:**

In the n-step REINFORCE, how is the return R_{t-j}^b computed? If it's over n steps, it means the RL agent basically ignores future rewards after n steps. In a general RL problem, this may lead to bad performance, but I guess for a local search method like in this paper, this may not be important. Could you comment on that point?

What did you choose to compare with Tabu search?
What are the state-of-the-art methods for the job shop scheduling problem? How does the proposed method compared to them?

---

> ### Author Response · Authors · 2023-11-19
> **Response to Reviewer jCmi**
>
> **C1: Page 1: "given that the topological relationships could be more naturally modelled among operations" is unclear.**
>
> **A**: A disjunctive graph effectively represents solutions by using directed arrows to indicate the order of processing operations, with each arrow pointing from a preceding operation to its subsequent one. The primary purpose of the disjunctive graph is to encapsulate the topological relationships among the nodes, which are defined by these directional connections. Thus, the disjunctive graph excels in illustrating the inter-dependencies between operations, in contrast to other visualization methods, such as Gantt charts, which are better suited for presenting explicit scheduling timelines.
>
> In formulating partial solutions within a construction step, the agent must possess accurate work-in-progress information, which encompasses specifics such as the extant load on each machine and the current status of jobs in process. The challenge encountered in this scenario relates to the difficulty in identifying suitable entities within the disjunctive graph that can store and convey such information effectively. While it is conceivable to encode the load of a machine using numerical values, no discrete node within the disjunctive graph explicitly embodies this information. Instead, a machine's representation is implicit and discerned through the collective interpretation of the operations it processes, effectively forming what is termed a 'machine clique.' This indirect representation necessitates a more nuanced approach to ensuring that the agent obtains the necessary information to construct viable partial solutions.
>
> We will clarify this in the updated manuscript.
>
> **C2: Page 3: the notion of $N_5$ neighbourhood could be explained more formally (instead of the current text and the example).**
>
> **A**: Thanks for the suggestion. Kindly note that we have made the necessary revisions to the document. Please refer to the updated manuscript for details once uploaded.
>
> **C3: Section 4.2.1: The first two sentences seem to contradict themselves. What is meant by "machine predecessor" becomes only clear later on.**
>
> **A**: We have carefully double-checked the definition of neighbours and found no contradictions. To prevent any potential misunderstanding, we amend the first two sentences as follows:
>
> "For the disjunctive graph, we define a node $U$ as a neighbour of node $V$ if an arc points from $U$ to $V$. Therefore, the dummy operation $O_S$ has no neighbours **(since no nodes are pointing to it)**, and $O_T$ has $|\mathcal{J}|$ neighbours **(since every job's last operation is pointing to it)**."
>
> **C4: Page 7: a closing curly brace is missing for $c_V$.**
>
> **A**: We deeply appreciate your meticulous attention in identifying the error. The correction has been made. Please refer to the updated manuscript.

---

> ### Author Response · Authors · 2023-11-19
> **Response to Reviewer jCmi cnt.**
>
> **C5: In the n-step REINFORCE, how is the return $R_{t-j}^b$ computed?**
>
> **A**: According to the definition of our reward in Eq.(1), the return $R_{t-j}^b$ for steps from $t-j$ until $t$ of instance $b$ can be calculated as:
>
> $$R_{t-j}^b = \sum_{t'=t-j}^{t}r(s_{t'}, a_{t'}) = C_{max}(s_{t-j}) - C_{max}(s^*_t),$$
>
> (with $s_t^*$ being the best solution found till step $t$), which is the improvement against solution $s_{t-j}$. It should be noted that the incumbent solution $s^*_t$ until time $t$ will stay the same for any $n$ steps.
>
> Intuitively, we divide the objective of improving the initial solution over $K$ (e.g., $K=5000$) steps into $K//n$ small objectives. We gradually update our DRL agent's policy when making little progress for any $n$ steps. One can easily verify that the sum of $n$-step return $R_{t-j}^b$ and the return $R_{K}^b$ calculated at the end of improvement step $K$ are the same, i.e., $\sum R_{t-j}^b = R_{K}^b$. Therefore, we have considered the future reward in the proposed $n$-step REINFORCE algorithm.
>
> **C6: Why did you compare with Tabu search? What are the state-of-the-art methods for the job shop scheduling problem? How does the proposed method compare to them?**
>
> **A**: It is widely committed in the manufacturing scheduling research community that Tabu search is one of the most promising metaheuristic methods for solving the JSSP [AR1] with state-of-the-art performance. Therefore, we regard the Tabu search as a strong baseline to compare against our method.
>
> [AR1] Xiong, Hegen, et al. "A survey of job shop scheduling problem: The types and models." *Computers \& Operations Research* 142 (2022): 105731.
>
> As mentioned in the paper, our aim is not to compete with SOTA metaheuristics but to present a framework that can learn local operators for JSSP that are stronger than hand-crafted ones, which can be potentially integrated into more complicated meta-heuristics to boost their performance further. Nonetheless, we have implicitly compared many SOTA metaheuristic methods. The relative gap to the best upper bound found in previous related works (the results in Table 1) is usually obtained with SOTA metaheuristic methods (The best-known results for these public benchmarks are available at http://optimizizer.com/TA.php and http://jobshop.jjvh.nl/), for example, an advanced memetic algorithm [AR2].
>
> [AR2] Constantino, O. H., \& Segura, C. (2022). A parallel memetic algorithm with explicit management of diversity for the job shop scheduling problem. *Applied Intelligence*, 52(1), 141-153.
>
> From the results in Table 1, our method achieved comparable performance (with 5.96\% average optimal gap) to the SOTA metaheuristics across 26 datasets from different benchmarks of various sizes. Furthermore, it is pertinent to underscore that, in contrast to state-of-the-art (SOTA) metaheuristic approaches that are intensely fine-tuned for specific problem instances, our method presents the **zero-shot** generalization capabilities. Notably, we demonstrate that our approach maintains a linear computational complexity with respect to problem size, as rigorously established in Theorem 1.

---

### Official Review · Reviewer_Sv6i · 2023-11-10

**Soundness:** 3 good
**Presentation:** 3 good
**Contribution:** 2 fair
**Rating:** 6
**Confidence:** 3

**Summary:**

This articled studied the problem of solving Job-shop scheduling problems using deep reinforcement learning focusing on heuristics. The authors have proposed a novel deep reinforcement learning based improvement heuristics where the graph representations are employed to encode complete solutions.

**Strengths:**

Good and decent review of literature based introduction gives a good readability.
Clear articulation of the MDP process with an example of state transition
Comparative study with Tabu Search algorithm for solving the instance
The computation complexity of the proposed method is linear with respect to the number of jobs and number of machines.

**Weaknesses:**

There are many studies that job shop scheduling that has been modeled as a MDP but very few citations are present in the article.

**Questions:**

how does the transition state in MDP (section 4.1) is calculated? need to explore in detail on how the transition space is updated with the current status information of all jobs and machines.
why the reward function goal is to improve the initial solutions? isn't closely correspond to scheduling goal?
How does the generalization capability of the agents are addressed in this article?
One way is to introduce order swapping mechanism with an instance and evaluate with another instance of the same size.
OpenAI Gym toolkit and associated kits provide reinforcement learning environment API - any exploration on this would have yielded results that could also be validated

---

> ### Author Response · Authors · 2023-11-19
> **Response to Reviewer Sv6i**
>
> **C1: There are many studies that job shop scheduling that has been modelled as a MDP but very few citations are present in the article.**
>
> **A**: Thank you for noticing that our article doesn't have many references on treating job shop scheduling as a Markov decision process (MDP). Following your suggestion, we have performed an exhaustive literature review and accordingly expanded the reference list in our manuscript to provide a more comprehensive analysis of the modelling approaches. The updated manuscript reflects these enhancements, with added references and insights marked in red.
>
> **C2: How is the MDP transition state (section 4.1) calculated? Need to explore in detail how the transition space is updated with the current status information of all jobs and machines.**
>
> **A**: We thank the reviewer for posing this question to help us further elucidate this state transition process, which we believe will benefit all the audiences in the future. All the contents will be updated in the appendix with a diagram to help readers better understand the state transition mechanism in this paper. Now, we provide the details on how the state is transited by giving concrete examples:
>
> The state transition is highly related to the $N_5$ neighbourhood structure. As we described in Preliminaries (Section 3), once our DRL agent selects the operation pair, say $(O_{13}, O_{21})$ in the sub-figure on the right of Figure 1, the $N_5$ operator will swap the processing order of  $O_{13}$ and $ O_{21}$. Originally, the processing order between $O_{13}$ and $O_{21}$ is $O_{13} \rightarrow O_{21}$. After swapping, the processing order will become $O_{13} \leftarrow O_{21}$. However, solely changing the orientation of arc $O_{13} \rightarrow O_{21}$ is not enough. In this case, we must break the arc $O_{21} \rightarrow O_{32}$ and introduce a new disjunctive arc $O_{13} \rightarrow O_{32}$ since the operation $O_{32}$ is now the last operation to be processed on the blue machine. In some cases, the state transition involves several steps. For example, given a processing order on some machine $O_{1} \rightarrow O_{2} \rightarrow O_3 \rightarrow O_4$, if we swap $O_{2}$ and $O_{3}$ (when $O_{2}$ and $O_{3}$ are the first/last two operations of some critical block), we need to break disjunctive arcs $O_{1} \rightarrow O_{2}$ and $O_3 \rightarrow O_4$, flip the orientation of arc $O_{2} \rightarrow O_{3}$ as $O_{2} \leftarrow O_{3}$, finally, add new arcs $O_3 \rightarrow O_1$ and $O_2 \rightarrow O_4$. So, the path after the transition will be $O_{1} \rightarrow O_{3} \rightarrow O_2 \rightarrow O_4$.
>
> In summary, after the transition, the new disjunctive graph (state) will have a different processing order for the machine assigned to the selected operation pair, but other machines stay unaffected.
>
> Then, the starting time (schedule) of every operation is re-calculated according to the proposed message-passing evaluator (Theorem 4.2).

---

> ### Author Response · Authors · 2023-11-19
> **Response to Reviewer Sv6i cnt.**
>
> **C3: Why is the reward function goal to improve the initial solutions? Doesn't it closely correspond to the scheduling goal?**
>
> **A**: In this paper, our aim is to learn a policy that improves an initial solution as much as possible. The improvement is defined according to specific scheduling objectives (e.g., minimizing makespan as in our paper). For a given initial solution, improving it as much as possible means optimizing the objective as much as possible. Therefore, our reward function directly corresponds to the scheduling goal.
>
> **C4: How are the generalization capabilities of the agents addressed in this article?**
>
> **A**: For the experiment setup, our model is trained with synthetic datasets generated based on the Talliard method (Section 5.1). Other benchmarking datasets, i.e., ABZ, FT, LA, SWV, ORB, and YN, are generated by following different distributions (please refer to Taillard, 1993; Adams et al., 1988; Fisher, 1963; Lawrence, 1984; Storer et al., 1992; Applegate \& Cook, 1991; Yamada \& Nakano, 1992, for details of the generation process of the public benchmarks). Therefore, for most results in Table 1, we have actually reported the **zero-shot** generalization performance of our method. The exceptions in Table 1 are the Taillard benchmark instances of the same size as in training, including 15$\times$15 and 20$\times$15. However, the generated training instances could be easier than those in the Taillard benchmarking datasets even though they follow the same generation procedure since only the most challenging instances are qualified for Taillard's benchmarking datasets (Taillard, 1993). Hence, we have also evaluated our model on different instances from the same distribution.
>
> We can explain the good zero-shot generalization ability of our model from the perspective of local isomorphism in graphs:
> Disjunctive graphs of different sizes are structurally similar in that each node has two parent nodes (a precedent constraint parent and a machine clique parent), and each node has corresponding two child nodes (a precedent constraint child and a machine clique child). Such a local topological structure remains constant regardless of the size of the graph. Therefore, concerning k-hop convolutions, the aggregated features and local topological information learned for the points are similar. Based on this observation, we have proposed CAM + TPM, where TPM is specifically responsible for extracting features of the graph's topological structure, thus allowing the topological features learned on small-scale graphs to generalize to larger-scale graphs.
>
> **C5: OpenAI Gym toolkit and associated kits provide reinforcement learning environment API - any exploration on this would have yielded results that could also be validated**
>
> **A**: The implementation of the proposed MDP (Markov Decision Process) in our paper follows the convention of OpenAI Gym, with interfaces like `init()`,`step()`, `reward()`, etc. We will also make our MDP modelling public, integrating it as an independent environment into the OpenAI Gym API. This will make it convenient for everyone to use and also facilitate others to replicate and validate the results presented in our paper.

---

> > ### Comment · Reviewer_Sv6i · 2023-11-22
> >
> > Thank you Authors for addressing my questions as well as other reviewers questions to improve the earlier version of the paper. My concerns and questions are answered and addressed in the latest revision of the article. Thank you!

---

### Author Response · Authors · 2023-11-19
**General Response**

First and foremost, we extend our heartfelt gratitude to the four reviewers for their diligent efforts. The reviewers' deep and intriguing inquiries have spurred substantial reflection on our part. For detailed responses to each of the questions posed by the reviewers, please refer to the information below.

We have revised the manuscript to reflect the feedback received, incorporating changes highlighted in red for clarity. The finalized version of the amended manuscript will be uploaded shortly. For now, we have prepared comprehensive responses to each comment for the purpose of facilitating discussion.

---

> ### Author Response · Authors · 2023-11-21
> **General Response Updated**
>
> Dear Reviewers,
>
> Please be informed that the revised version of the manuscript has been uploaded for your convenience. We remain committed to further refining it as necessary up until the point of publication.
>
> The authors

---

> ### Author Response · Authors · 2023-11-22
> **Revert to the Initially Submitted Nine-Page Manuscript**
>
> Dear Esteemed Reviewers,
>
> We wish to convey that the manuscript, which had previously been updated, has now been reverted to its original nine-page format. The stipulations regarding page limits applicable during the rebuttal phase have guided this decision.
>
> We assure you that, despite this adjustment, our responses within the manuscript remain thorough and unequivocally clear to support and encourage detailed discussions.
>
> We kindly ask for your understanding regarding these modifications, and we sincerely apologise for any inconvenience this may have caused. Please be assured that the pertinent changes will be diligently incorporated into the final version of the manuscript.
>
> We appreciate your attention and look forward to your invaluable feedback.
>
> Warm regards,
> The Authors

---

### Meta-Review · Area_Chair_GmMp · 2023-12-06

**Metareview:**

The paper studies the job-shop scheduling problem and proposes a deep RL approach to approximately solve them. The key contributions are a novel graph NN representation of the solution space, and a novel mechanism to efficiently evaluate proposed solutions. All of the reviewers agree that the paper convincingly outperforms deep RL baselines and other optimization meta-heuristics.

**Justification For Why Not Higher Score:**

Reviewers raised concerns that the evaluation is narrowly focused on a specific kind of scheduling problems. During the author feedback phase, they ran an additional experiment on a variant of the JSSP problem to demonstrate that the method can be more generally applied. Applying the method to a broader class of problems can increase the significance of the result to beyond the subset of the audience interested in core scheduling problems.

**Justification For Why Not Lower Score:**

All of the reviewers agree that the paper makes novel contributions, is written clearly, advances the state of the art of ML for a subclass of scheduling problems, and is above the bar for publication.

---

### Decision · Program_Chairs · 2024-01-16

Accept (poster)